# Targeting an anchored phosphatase-deacetylase unit restores renal ciliary homeostasis

Janani Gopalan[1], Mitchell H Omar[1], Ankita Roy[2,3], Nelly M Cruz[2,3], Jerome Falcone[1], Kiana N Jones[1], Katherine A Forbush[1], Jonathan Himmelfarb[2], Benjamin S Freedman[2,3], John D Scott[1]*

[1]Department of Pharmacology, University of Washington, Seattle, United States; [2]Kidney Research Institute, Division of Nephrology, Department of Laboratory Medicine and Pathology, University of Washington, Seattle, United States; [3]Institute for Stem Cell and Regenerative Medicine, University of Washington, Seattle, United States

*For correspondence:
scottjdw@uw.edu

Competing interests: The authors declare that no competing interests exist.

**Abstract** Pathophysiological defects in water homeostasis can lead to renal failure. Likewise, common genetic disorders associated with abnormal cytoskeletal dynamics in the kidney collecting ducts and perturbed calcium and cAMP signaling in the ciliary compartment contribute to chronic kidney failure. We show that collecting ducts in mice lacking the A-Kinase anchoring protein AKAP220 exhibit enhanced development of primary cilia. Mechanistic studies reveal that AKAP220-associated protein phosphatase 1 (PP1) mediates this phenotype by promoting changes in the stability of histone deacetylase 6 (HDAC6) with concomitant defects in actin dynamics. This proceeds through a previously unrecognized adaptor function for PP1 as all ciliogenesis and cytoskeletal phenotypes are recapitulated in mIMCD3 knock-in cells expressing a phosphatase-targeting defective AKAP220-ΔPP1 mutant. Pharmacological blocking of local HDAC6 activity alters cilia development and reduces cystogenesis in kidney-on-chip and organoid models. These findings identify the AKAP220-PPI-HDAC6 pathway as a key effector in primary cilia development.

## Introduction

Kidneys recycle about 180 liters of fluid every day to partition nutrients and remove toxins from blood (*Saborio et al., 2000*). Water reabsorption from luminal fluid is triggered by the hormone arginine vasopressin via phosphorylation-dependent translocation of aquaporin-2 water pores to the apical surface of kidney collecting ducts (*Bankir et al., 2013*; *Noda et al., 2010*; *Yui et al., 2012*). Not surprisingly, defects in renal water homeostasis have pathophysiological consequences. Approximately 35 million Americans suffer from chronic kidney diseases, characterized as a gradual loss of renal function (*Hemmelgarn et al., 2006*). Polycystic kidney diseases are disorders where the collecting ducts become enlarged with fluid filled cysts that reduce glomerular filtration rate (*Wilson, 2004*). Autosomal dominant polycystic kidney disease (ADPKD) with an estimated prevalence of 1 in 600 people, is a common genetic disorder associated with end-stage renal failure (*Halvorson et al., 2010*). Clinical evidence indicates that primary cilia function is altered in ADPKD (*Ma et al., 2013*; *Ma et al., 2017*). Hence this chronic renal disorder is classified as a ciliopathy (*Badano et al., 2006*; *Fliegauf et al., 2007*).

The primary cilium is a microtubule-based organelle protruding from the surface of most mammalian cells (*Satir et al., 2010*). In the kidney, primary cilia respond to fluctuations in fluid-flow through collecting ducts. They convert mechanical stimuli into biochemical signals to elicit developmental and regulatory responses (*Mukhopadhyay et al., 2013*; *Somatilaka et al., 2020*). Disease causing

mutations in the ciliary transmembrane proteins polycystin 1 (*PKD1*) and polycystin 2 (*PKD2*) underlie ADPKD (*Hughes et al., 1995*; *Mochizuki et al., 1996*). Both proteins are components of a receptor-channel complex that responds to local second messenger signals (*Harris and Torres, 2014*). Accordingly, a ciliary hypothesis has been formulated that implicates defective calcium and cAMP signaling in the ciliary compartment as a factor in disease progression (*Ma et al., 2017*; *Winyard and Jenkins, 2011*).

A-kinase anchoring proteins (AKAPs) spatially constrain second messenger regulated kinases, protein phosphatases, and GTPase effector proteins within subcellular compartments (*Bucko and Scott, 2021*; *Langeberg and Scott, 2015*; *Omar and Scott, 2020*; *Smith et al., 2017*; *Smith et al., 2018*; *Taskén and Aandahl, 2004*; *Whiting et al., 2015*). Several AKAPs participate in renal signaling, yet only a few anchoring proteins reside in cilia (*Choi et al., 2011*; *Gopalan et al., 2021*; *Jo et al., 2001*; *May et al., 2020*; *Stefan et al., 2007*). Ciliary AKAPs are postulated to position protein kinase A (PKA) as a negative regulator of hedgehog signaling (*Breslow et al., 2018*; *Mukhopadhyay et al., 2013*; *Somatilaka et al., 2020*). AKAP220 is a multifunctional anchoring protein that sequesters PKA, GSK3, the Rho GTPase effector IQGAP-1, and protein phosphatase 1 (PP1) (*Logue et al., 2011*; *Schillace and Scott, 1999*; *Whiting et al., 2015*). Each AKAP220-binding partner is implicated in local signaling events that potentiate ADPKD. PKA and PP1 bi-directionally control signaling in the ciliary compartment, whereas reduced Rho GTPase activity contributes to expansion of renal cysts (*Parnell et al., 2012*; *Ye et al., 2017*).

Here, we show that AKAP220KO mice exhibit enhanced development of primary cilia. Mechanistic studies reveal that AKAP220 associated PP1 drives this phenotype by promoting changes in actin dynamics and histone deacetylase 6 (HDAC6) stability. Changes in local HDAC6 activity alters cilia development and reduces cystogenesis in cellular models of ADPKD. These findings point toward the AKAP220-PPI-HDAC6 pathway as a signaling center for the control of ciliary development and a potential target for the HDAC6 inhibitor drug tubacin.

## Results

### Loss of AKAP220 promotes renal cilia assembly

AKAP220 modulates cytoskeletal signaling events through its ability to recruit kinases, phosphatases and the small GTPase effector protein, IQGAP1 (*Figure 1A*; *Logue et al., 2011*). AKAP220KO mice display mild defects in water homeostasis and altered aquaporin 2 (AQP2) trafficking that are linked to disruption of an apical actin barrier in the kidney collecting ducts (*Figure 1B*; *Whiting et al., 2016*). Further investigation in kidney sections from wild-type and AKAP220KO mice led to the unexpected discovery that deletion of this anchoring protein correlated with increased numbers of primary cilia decorating each collecting duct (*Figure 1C–H*). The GTPase Arl13b (red) served as a ciliary marker, staining of AQP2 (green) marked kidney collecting ducts and DAPI (blue) highlighted nuclei (*Figure 1C & D*). Analysis of tissue sections collected from several animals are presented in *Figure 1I*, measuring a 3.5-fold increase in the percentage of ciliated cells in AKAP220KO, as compared to wild-type. Additional measurements determined that cilia in AKAP220KO collecting ducts are 1.68-fold longer than their wild-type counterparts (1.7 µm vs 1.02 µm; *Figure 1J*).

Independent validation of this observation was provided when CRISPR-Cas9 gene-editing was used to delete AKAP220 from mouse Inner Medullary Collecting Duct (mIMCD3) cells. Immunoblot analysis confirmed the loss of AKAP220 in two independent clones (*Figure 1K*, top panel, lanes 2 and 3). GAPDH was used as loading control (*Figure 1K*, bottom panel). Immunofluorescent detection of primary cilia in each AKAP220KO clone measured an 8- to 10-fold increase in the percentage of ciliated cells in comparison to wild-type mIMCD3 cells (*Figure 1L–O*). Quantification is presented in *Figure 1O*. Serum starvation is typically used to ciliate mIMCD3 cells in culture (*Westlake et al., 2011*). However, serum starvation of AKAP220KO cell lines enhanced detection of multi-ciliated cells (*Figure 1—figure supplement 1A–G*). This made single cell quantification less accurate. For this reason, all further analyzes were conducted in asynchronous cultures. Rescue experiments allowed us to attribute the increases in cilia to loss of AKAP220 (*Figure 1P–U*). AKAP220KO cells were transfected with a GFP-tagged AKAP220 construct (*Figure 1P*). Immunofluorescent detection of Arl13b (red) detected cilia and GFP-fluorescence confirmed rescue with the anchoring protein (*Figure 1Q and R*). GFP served as a control (*Figure 1S and T*). Quantification from three independent experiments

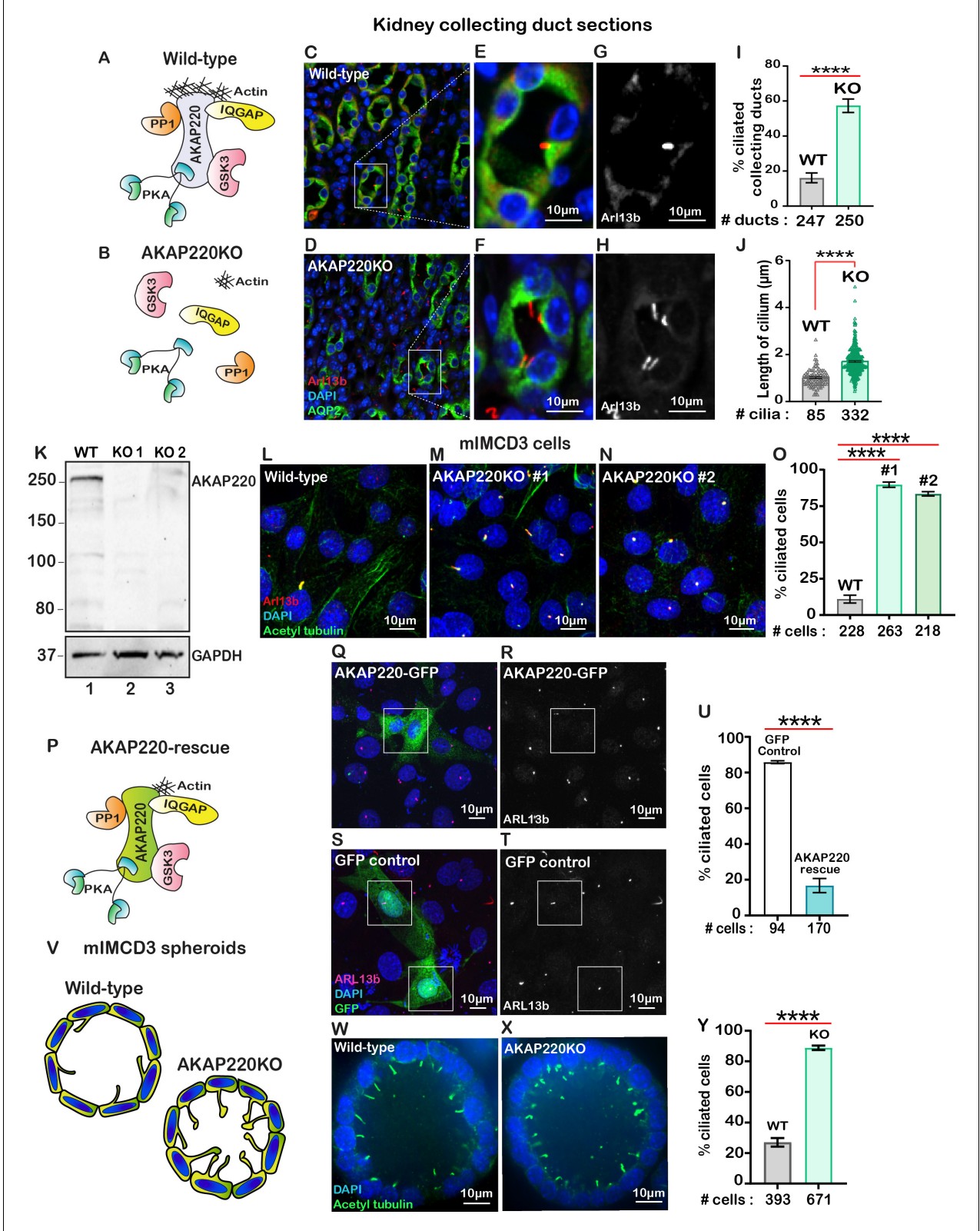

**Figure 1.** Loss of AKAP220 enhances ciliogenesis. (**A**) Schematic of AKAP220 interaction with selected binding partners. (**B**) Disruption of this signaling complex upon removal of AKAP220. Protein kinase A (blue), Glycogen synthase kinase-3 (pink), Protein phosphatase 1 (orange) and IQGAP (yellow) are indicated. (**C–H**) Immunofluorescent staining of kidney collecting ducts with Arl13b (red), Aquaporin-2 (green) and DAPI (blue) from (**C**) wild-type and (**D**) AKAP220KO mice. (**E and F**) Enlarged sections from wild-type and AKAP220KO mice. (**G and H**) Gray scale images of Arl13b. (**I**) Quantification (%

*Figure 1 continued on next page*

*Figure 1 continued*

ciliated collecting ducts) in wild-type (gray bar) and AKAP220KO (green bar). ****p<0.0001. (J) Quantification of cilia length (μm) in wild-type (gray bar) and AKAP220KO (green bar). ****p<0.0001. Crispr-Cas9 gene editing of AKAP220 in mIMCD3 cells. (K) Immunoblot detection of AKAP220 (top) and GAPDH loading control (bottom) from wild-type (lane 1) and AKAP220KO (lane 2) cell lysates. (L–N) Immunofluorescent detection of primary cilia with acetyl tubulin (green), Arl13b (red) and DAPI (blue) in wild-type, and two independent clones of AKAP220KO mIMCD3 cells. (O) Quantification (% ciliated cells) from wild-type (gray column), AKAP220KO#1 (green column) and AKAP220KO#2 (dark green column). ****p<0.0001, N=3. (P) Schematic depicting reformation of the signaling complex upon rescue with AKAP220. Immunofluorescent detection of Arl13b (pink), GFP (green) and DAPI (blue) in (Q) pEGFP-AKAP220 or (S) GFP-control transfected AKAP220KO mIMCD3 cells. Gray scale image of Arl13b in (R) control cells and (T) AKAP220-rescued cells. (U) Quantification (% ciliated cells) in pEGFP-AKAP220 (black bar) or GFP-control cells (teal bar). ****p<0.0001, N=3. (V) Schematic of wild-type and AKAP220KO mIMCD3 spheroids. Immunofluorescent staining with acetyl tubulin (green) and DAPI (blue) in (W) wild-type and (X) AKAP220KO spheroids. (Y) Quantification (% ciliated cells) in wild-type (gray column) and AKAP220KO (green column) spheroids. ****p<0.0001, N=3. All error bars are s.e.m. p Values were calculated by unpaired two-tailed Student's t-test. Scale bars (10 μm). Number of cells analyzed indicated below each column.

The online version of this article includes the following source data and figure supplement(s) for figure 1:

**Source data 1.** Percent ciliated collecting ducts in kidney sections.
**Source data 2.** Length of primary cilia in kidney sections.
**Source data 3.** Rescue of GFP-AKAP220 in AKAP220KO mIMCD3 cells.
**Source data 4.** Percent ciliated cells measured in spheroids.
**Figure supplement 1.** Quantification of cilia number in gene-edited mIMCD3 cells.
**Figure supplement 2.** Deletion of AKAP150 has no effect on primary cilia development.
**Figure supplement 3.** Sequencing data for CRISPR-Cas9 gene edited mIMCD3 cells.

confirmed that rescue with AKAP220 dramatically reduced the number of ciliated cells as compared to rescue with GFP (*Figure 1U*).

Since another anchoring protein AKAP150 has been detected in primary cilia, it was important to establish if this AKAP also contributed to primary cilia development (*Choi et al., 2011*). Gene-editing was used to generate mIMCD3 cells lacking AKAP150. Double knockout cells lacking AKAP150 and AKAP220 were also produced. Immunofluorescent detection of ARl13b (green) and acetyl-tubulin (red) as ciliary markers revealed that the loss of AKAP150 alone had no effect on cilia development (*Figure 1—figure supplement 2A & B*). Yet, a pronounced increase in the ciliated population was observed in AKAP220/150 double knockout cells (*Figure 1—figure supplement 2C & D*). These additional studies indicate that AKAP150 signaling does not support renal ciliogenesis.

Renal spheroids recapitulate features of ductal architecture for disease modeling (*Giles et al., 2014*; *Figure 1V*). We cultured mIMCD3 cells in Matrigel for 72 hr to form three-dimensional spheroids (*Figure 1W & X*). Immunofluorescent detection of the ciliary marker acetyl-tubulin (green) confirmed that loss of AKAP220 correlated with persistence of primary cilia in mIMCD3 KO spheroids. Detection of DAPI (blue) marked nuclei (*Figure 1W & X*). Thus, local signaling mechanisms that proceed through AKAP220 impact ciliary development in cellular, three-dimensional spheroid and animal models of the renal microenvironment.

## AKAP220 contributes to histone deacetylase six stability

Primary cilia formation requires tubulin heterodimers that are stabilized by acetylation on lysine 40 (*Portran et al., 2017*). This prompted us to consider whether acetylated tubulin was abundant in cells lacking pAKAP220. Experiments were conducted in two phases. First, immunofluorescent detection of acetylated alpha tubulin confirmed that this modified form was prevalent in AKAP220KO cells (*Figure 2A & B*). Second, immunoblot analysis showed that acetyl tubulin was more prominent in AKAP220KO cell lysates as compared to wild-type (*Figure 2C*, top panel). α-tubulin was consistent in both cell lines (*Figure 2C*, mid panel). GAPDH was a loading control (*Figure 2C*, lower panel). Amalgamated data from four independent experiments are quantified in *Figure 2D*.

Histone deacetylase 6 (HDAC6) catalyzes the de-acetylation of tubulin to promote depolymerization of primary cilia (*Hubbert et al., 2002*; *Ran et al., 2015*). Hence, we reasoned that the enhanced detection of acetylated tubulin in AKAP220KO cells may be a consequence of reduced HDAC6 activity. Five complementary methods tested this postulate. First, immunofluorescent detection of HDAC6 (cyan) confirmed its enhanced compartmentalization at the base of primary cilia in wild-type mIMCD3 cells (*Figure 2E*; *Figure 2—figure supplement 1A*). Conversely, the subcellular

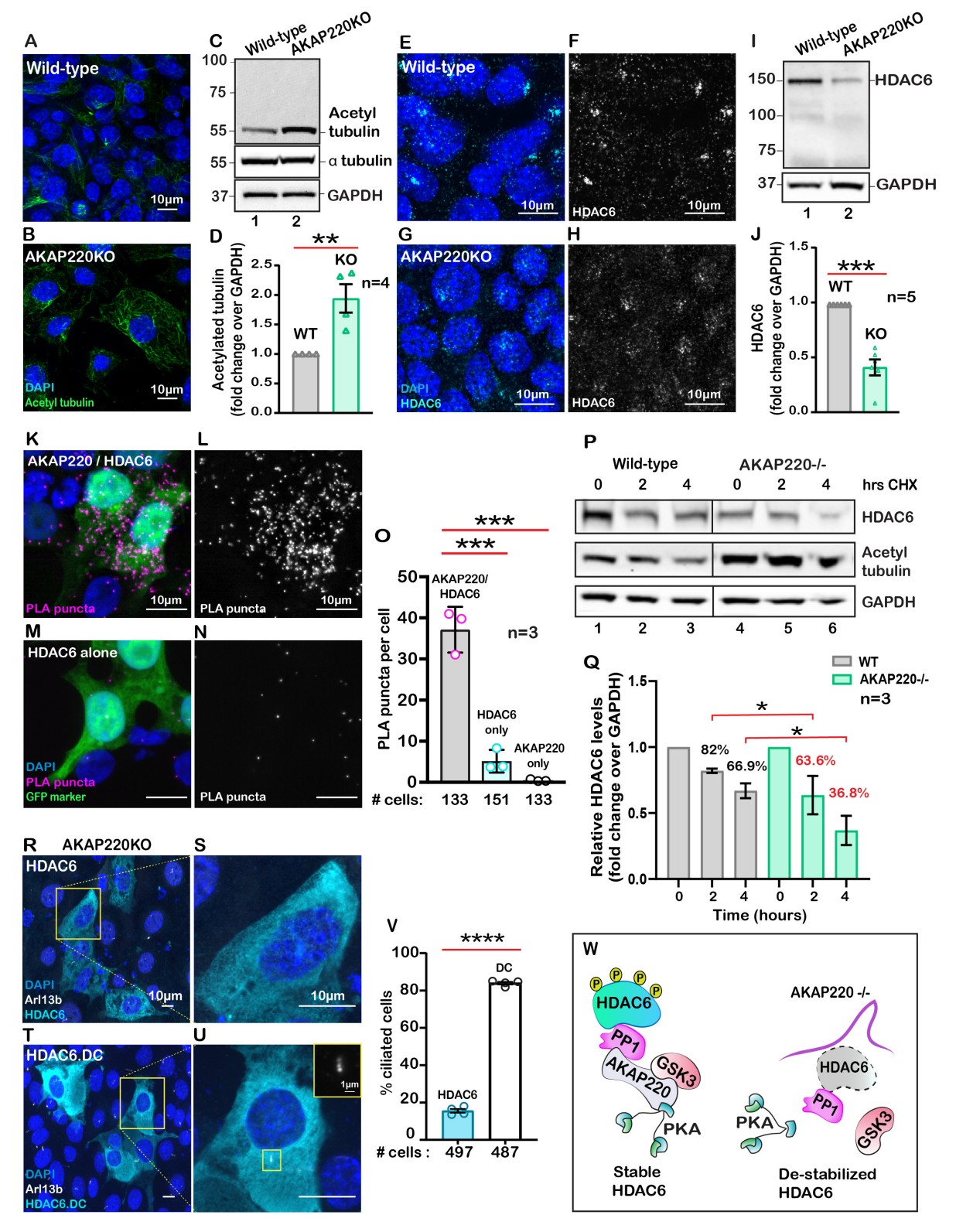

**Figure 2.** AKAP220 influences tubulin deacetylation. Immunofluorescent detection of acetyl tubulin (green) and DAPI (blue) in (A) wild-type and (B) AKAP220KO mIMCD3 cells. (C) Immunoblot detection of acetylated tubulin (top), alpha tubulin (mid) and GAPDH loading control (bottom), in wild-type (lane 1) and AKAP220KO (lane 2) cell lysates. (D) Quantification by densitometry of acetylated tubulin in wild-type (gray column) and AKAP220KO (green column) lysates. **p<0.01, N=4. Immunofluorescent staining of HDAC6 (cyan) and DAPI (blue) in (E) wild-type and (G) AKAP220KO cells. Gray scale

*Figure 2 continued on next page*

*Figure 2 continued*

images of HDAC6 in (**F**) wild-type and (**H**) AKAP220KO cells. (**I**) Immunoblot detection of HDAC6 (top) and GAPDH as loading control (bottom) in wild-type (lane 1) and AKAP220KO (lane 2) cell lysates. (**J**) Quantification by densitometry of HDAC6 in wild-type (gray column) and AKAP220KO (green column) lysates. ***p<0.001, N=5. (**K**) Proximity ligation (PLA) detection of V5-AKAP220/HDAC6 subcomplexes (pink), DAPI (blue) in cells expressing GFP (green) as a transfection marker. (**L**) Gray scale image highlights V5-AKAP220/HDAC6 PLA puncta. (**M and N**) Control PLA experiments in cells treated with anti-HDAC6 antibody alone. (**O**) Amalgamated data (PLA puncta/cell) from three independent experiments is presented. Cycloheximide pulse-chase assay investigated HDAC6 stability. (**P**) Immunoblot of HDAC6 (top), acetylated tubulin (mid) and loading control GAPDH (bottom) from wild-type (lanes 1–3) and AKAP220KO (lanes 4–6) from mIMCD3 cells treated with cycloheximide. Data collected over a time course (0–4 hr). (**Q**) Quantification of amalgamated data by densitometry (N=3). HDAC6 levels in wild-type (gray) and AKAP220KO (green) are indicated. Levels of protein (%) are normalized to wild-type control (no treatment, 0 hr). (**R-V**) AKAP220KO mIMCD3 cells were transfected with (**R**) flag-HDAC6 or (**T**) catalytically inactive mutant HDAC6.DC. Immunofluorescent detection of HDAC6 (cyan), Arl13b (white) and DAPI (blue). Enlarged sections depict (**S**) loss of primary cilium upon overexpression of active HDAC6 and (**U**) intact cilium upon overexpression of inactive mutant HDAC6.DC. (**V**) Quantification (% ciliated cells) in AKAP220KO cells transfected with flag-HDAC6 (black column) and HDAC6.DC (cyan column). ****p<0.0001, N=3. (**W**) Schematic of how recruitment to the AKAP220-signaling complex stabilizes HDAC6 through local phosphorylation. All error bars are s.e.m. p Values were calculated by unpaired two-tailed Student's t-test. Scale bars (10 µm). Number of cells analyzed indicated below each column.

The online version of this article includes the following source data and figure supplement(s) for figure 2:

**Source data 1.** Densitometry of acetylated tubulin western blots.
**Source data 2.** Densitometry of HDAC6 western blots.
**Source data 3.** Densitometry of HDAC6 in cycloheximide-chase experiments.
**Source data 4.** Percent ciliated AKAP220KO cells transfected with Flag-HDAC6 or HDAC6.DC.
**Source data 5.** Quantification of puncta in proximity ligation assay.
**Figure supplement 1.** Further characterization of AKAP220-HDAC6 interface.
**Figure supplement 2.** Additional Proximity ligation (PLA) controls.

distribution of HDAC6 was altered in AKAP220KO mIMCD3 cells (*Figure 2G*; *Figure 2—figure supplement 1C*). This is most apparent in the gray scale images representing HDAC6 alone (*Figure 2F & H*, *Figure 2—figure supplement 1B & D*). Second, immunoblot analysis of HDAC6 protein in mIMCD3 cell lysates detected reduced levels of the deacetylase in cells that lack AKAP220, as compared to wild-type controls (*Figure 2I*, top panel, lane 2). GAPDH served as a loading control (*Figure 2I*, bottom panel). Amalgamated data from five independent experiments indicated that deletion of AKAP220 correlated with a 50% reduction of total HDAC6 in mIMCD3 cells (*Figure 2J*). This argues that interface with AKAP220 serves to stabilize HDAC6.

Third, proximity ligation assay (PLA) detected AKAP220-HDAC6 complexes in situ. This approach identifies protein-protein interactions that occur within a range of 40–60 nm (*Whiting et al., 2015*). PLA puncta indicative of AKAP220/HDAC6 complexes were detected in cells transfected with V5-tagged AKAP220 (*Figure 2K*). Gray scale image emphasizes the distribution of AKAP220/HDAC6 complexes (*Figure 2L*). In contrast, PLA puncta were reduced in control experiments performed in untransfected cells or in the absence of HDAC6 or V5 antibodies (*Figure 2M & N*, Sup *Figure 2E–I*). GFP was a transfection marker. Quantification from three distinct experiments is presented (*Figure 2O*) and additional controls are included in *Figure 2—figure supplement 2A–E*.

Fourth, we performed cycloheximide-chase experiments. Cell lysates were prepared at selected time points (0–4 hr; *Figure 2P*). Immunoblot detection of HDAC6 (top panel) and acetylated tubulin (mid panel) monitored the stability and activity of the de-acetylase over time. GAPDH served as a loading control (bottom panel). In wild-type mIMCD3 cells, HDAC6 protein was relatively constant and acetylated tubulin levels were low (*Figure 2P*, lanes 1–3). Conversely, upon loss of the anchoring protein, HDAC6 levels were markedly reduced over time and acetylated tubulin levels were elevated (*Figure 2P*, lanes 4–6). Densitometric analysis of HDAC6 levels in wild-type (gray) and AKAP220KO (green) from three independent experiments are presented (*Figure 2Q*). These results show a significant decrease in HDAC6 stability in the absence of the anchoring protein (*Figure 2P & Q*). Finally, overexpression of murine HDAC6 (cyan) abrogated primary cilia formation in AKAP220KO cells (*Figure 2R & S*). In contrast, expression of a catalytically inactive HDAC6 mutant (H216A, H611A) had no effect on cilia formation in the AKAP220 null background (*Figure 2T & U*). Amalgamated data from three experiments are presented (*Figure 2V*). Collectively, these studies implicate HDAC6 activity in AKAP220-mediated control of primary cilia development. Mechanistically, dephosphorylation of HDAC6 favors its degradation (*Ran et al., 2020*). Protein phosphatase 1(PP1) is a well-characterized binding partner of AKAP220 that also interacts with this deacetylase (*Brush et al., 2004*).

Therefore, we reasoned that PP1 may serve as an adaptor protein that bridges HDAC6 to AKAP220 at the ciliary compartment (*Figure 2W*).

## AKAP220-anchored protein phosphatase one alters primary cilia turnover

Protein phosphatase one interacts with AKAP220 primarily through its KVQF motif (*Schillace and Scott, 1999*) and has recently been implicated in trafficking of polycystin-1 to cilia (*Luo et al., 2019*). We used CRISPR-Cas9 gene editing to generate a knock-in cell line where the phosphatase-targeting motif was replaced by a TATA sequence (*Figure 3A & B*; *Figure 3—figure supplement 1A & B*). Co-immunoprecipitation assays confirmed that AKAP220-ΔPP1 is unable to anchor PP1 (*Figure 3—figure supplement 1C*). Immunoblot analysis confirmed expression of AKAP220-ΔPP1 as compared to knockout control (*Figure 3C*, top panel, lane 2). GAPDH was a loading control (*Figure 3C*, bottom panel). Next, we investigated if loss of the principal phosphatase targeting motif from AKAP220 alters its ability to interact with HDAC6. Immunoprecipitation of HDAC6 from wild-type mIMCD3 cells results in co-fractionation of AKAP220 (*Figure 3D*, lane 1). Importantly, co-fractionation of the anchoring protein is markedly reduced when experiments are performed in cells expressing AKAP220-ΔPP1 (*Figure 3D*, lane 2). Immunoblot detection of AKAP220 (top), HDAC6 (mid) in cell lysates and GAPDH loading controls (bottom) reveal equivalent levels of all proteins in mIMCD3 cell lysates (*Figure 3D*). These data indicate removal of the principle PP1-interaction motif on AKAP220 negatively impacts association with HDAC6.

HDAC6 is a phosphoprotein. The AKAP220-associated kinase GSK3 phosporylates HDAC6 on serine 22 (*Chen et al., 2010*). Thus recruitment into AKAP220 signaling islands would optimally position HDAC6 for control by reversible phosphorylation. We monitored the phosphoryaltion status of Ser22 on HDAC6 in wild-type and AKAP220-ΔPP1 cells (*Figure 3—figure supplement 2A & B*). Immunoblot detection of pSer22 HDAC6 was robust in wild-type mIMCD3 cell lysates (*Figure 3—figure supplement 2A*, upper panel, lane 1). Importantly, the p-HDAC6 signal was reduced in cell lysates from AKAP220-ΔPP1 mIMCD3 cells (*Figure 3—figure supplement 2A*, upper panel, lane 2). Control experiments confirmed equivalent amounts of total HDAC6 in cell lysates from both genotypes (*Figure 3—figure supplement 2B*, upper panel both lanes). Thus phospho-HDAC6 levels are reduced in AKAP220-ΔPP1.

Immunofluorescence detection of p-HDAC6 provided independent confirmation of this notion (*Figure 3—figure supplement 2C–I*). mIMCD3 cells were stained for pHDAC6 (red), AKAP220 (green). Acetyl tubulin (magenta) was used as a ciliary marker and DAPI (blue) marked nuclei (*Figure 3—figure supplement 2C & F*). In wild-type cells, co-distribution of AKAP220 and p-HDAC6 was detected at the base of cilia (*Figure 3—figure supplement 2D*, arrow). In contrast, the p-HDAC6 signal was reduced in AKAP220-ΔPP1 cells (*Figure 3—figure supplement 2G*, arrow). Quantification of p-HDAC6 levels at the base of cilia in multiple cells is presented (*Figure 3—figure supplement 2I*). These data are consistent with dissociation of HDAC6 from the anchored kinase-phosphatase signaling complex in a manner that promotes a concomitant loss of regulation via protein phosporylation.

Next, we investigated the impact of AKAP220-ΔPP1 on ciliogenesis. Immunofluorescence experiments revealed that loss of a mere four-residue motif that enables PP1-anchoring to AKAP220 correlated with dramatically increased cilia numbers (*Figure 3E & F*). Arl13b (red) and acetyl tubulin (green) were used as ciliary markers and nuclei were detected by DAPI (blue). This striking phenotype is clearly portrayed in the gray scale images of Arl13b (*Figure 3G & H*). Additionally, the differential width of the three-dimensional surface plots show that primary cilia morphology is altered in the AKAP220-ΔPP1 cells, as compared to wild-type controls (*Figure 3I & J*). Results from at least three independent experiments are quantified (*Figure 3O*). Parallel analyses established that acetyl-tubulin was elevated in AKAP220-ΔPP1 (*Figure 3K & L*). This is highlighted in insets and 3D surface plots of acetylated tubulin (*Figure 3M & N*). Collectively, these results indicate that PP1 anchoring to AKAP220 is necessary for regulation of ciliary development and its loss is sufficient to drive enhanced ciliation. We reasoned that PP1 may serve as an adaptor protein that incorporates histone deacetylase six into AKAP220 signaling complexes.

A fluorometric assay monitored HDAC6 activity in each cell type (*Figure 3P*). As expected, HDAC6 activity was reduced in AKAP220KO and AKAP220-ΔPP1, as compared to the wild-type mIMCD3 cells. Amalgamated data from four experiments are presented (*Figure 3P*). These findings

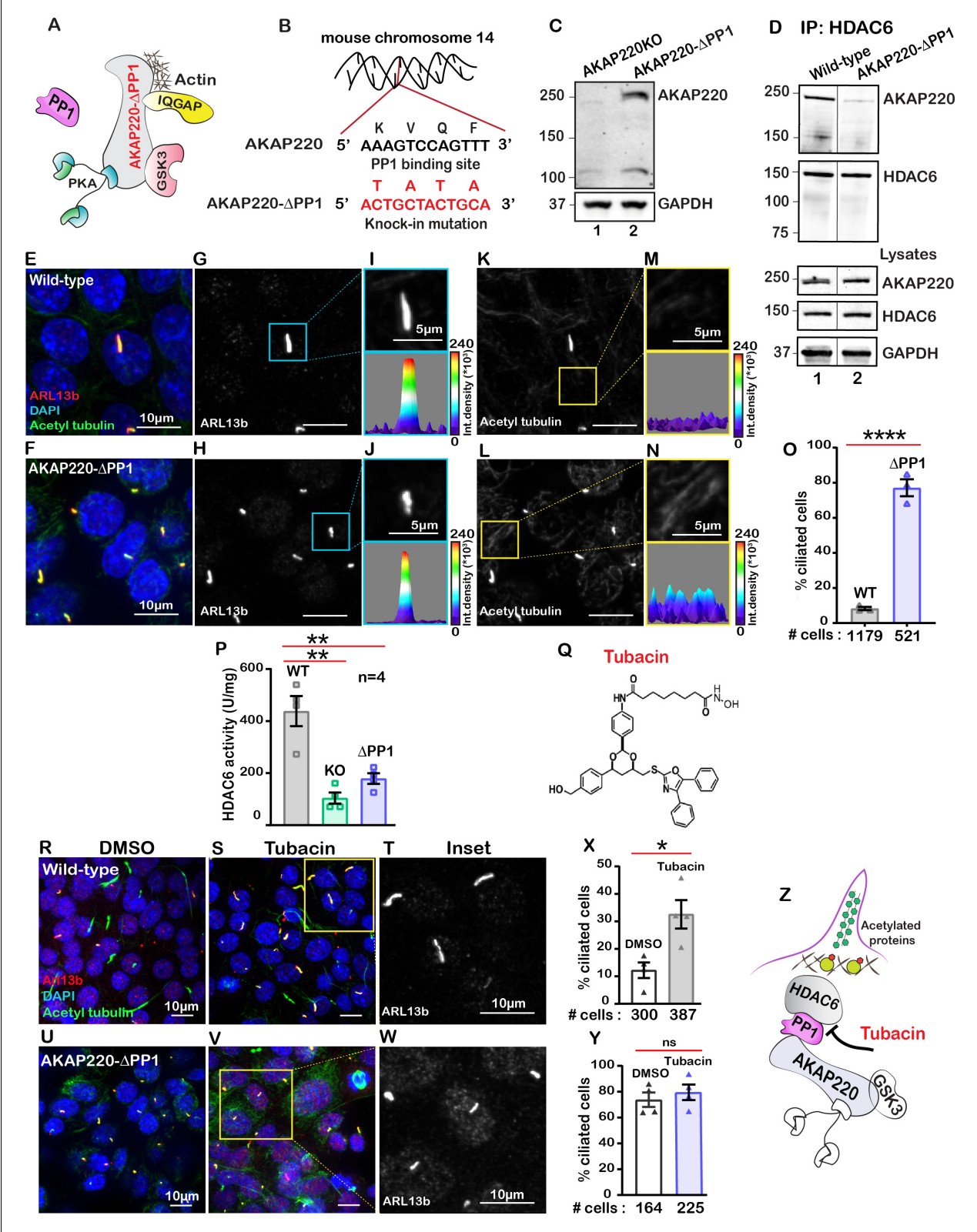

**Figure 3.** Anchored protein phosphatase one is necessary for HDAC6 activity. (**A**) Schematic of AKAP220-ΔPP1. Binding partners are indicated. Gene editing deleted the principal phosphatase binding site (KVQF) on AKAP220. (**B**) Nucleotide sequencing reveals substitution of the KVxF motif. (**C**) Immunoblot detection of AKAP220 (top) and GAPDH loading control (bottom) in AKAP220KO (lane 1) and AKAP220-ΔPP1 (lane 2) mIMCD3 cell lysates. (**D**) Loss of PP1-targeting motif in AKAP220 negatively impacts association with HDAC6. Co-immunoprecipitation studies show that wild-type AKAP220

*Figure 3 continued on next page*

*Figure 3 continued*

recruits HDAC6 (lane 1). AKAP220-ΔPP1 recruits less HDAC6 (lane 2). Immunoblot detection of AKAP220 (top), HDAC6 (mid) in cell lysates and GAPDH loading controls (bottom) reveal that equivalent levels of both proteins were present in mIMCD3 cell lysates. (E–O) Immunofluorescent detection of acetyl tubulin (green), Arl13b (red), and DAPI (blue) in (E) wild-type and (F) AKAP220-ΔPP1 cells. Gray scale images of Arl13b in (G) wild-type and (H) AKAP220-ΔPP1 cells. A single enlarged cilium (top) and corresponding three-dimensional surface plots (bottom) from (I) wild-type and (J) AKAP220-ΔPP1 cells. Gray scale images of acetylated tubulin in (K) wild-type and (L) AKAP220-ΔPP1 cells. Enlarged sections from (K and L) (top) and corresponding three-dimensional surface plots (bottom) from (M) wild-type and (N) AKAP220-ΔPP1 cells. (O) Quantification (% ciliated cells) in wild-type (gray) and AKAP220-ΔPP1 (blue). ****p<0.0001, N=3. (P) HDAC6 activity levels (A.U.) in wild-type (gray), AKAP220KO (green) and AKAP220-ΔPP1 (blue) cells as assessed by Bioline's activity assay. **p<0.01, N=4. (Q) Chemical structure of HDAC6 inhibitor tubacin. (R-Y) Tubacin enhances ciliogenesis in the presence of native AKAP220. Wild-type mIMCD3 cells treated with (R) DMSO or (S) tubacin (2 µM) for 4 hr. Immunofluorescent staining with acetyl tubulin (green), Arl13b (red), and DAPI (blue). (T) Higher magnification gray scale image of Arl13b staining. (X) Quantification (% ciliated cells) in DMSO (white) and tubacin-treated (gray) wild-type cells. *p<0.05, ns=non-significant; N=3. (U–Y) Tubacin has no effect on AKAP220-ΔPP1 cells. (U) DMSO and (V and W) tubacin-treated AKAP220-ΔPP1 cells. (Y) Quantification (% ciliated cells) and analysis as described above in DMSO (white) and tubacin-treated (blue). (Z) Schematic of proposed tubacin mechanism of action on AKAP220-signaling complex. All error bars are s.e.m. p Values were calculated by unpaired two-tailed Student's t-test. Scale bars (10 µm). Number of cells analyzed indicated below each column.

The online version of this article includes the following source data and figure supplement(s) for figure 3:

**Source data 1.** Percent ciliated DMSO or tubacin-treated mIMCD3 cells.
**Figure supplement 1.** Characterization of AKAP220-ΔPP1 cell line.
**Figure supplement 2.** Investigating if AKAP220-HDAC6 interaction is altered in the AKAP220-ΔPP1 cells.
**Figure supplement 3.** Serum starvation drives mutant cells to multiciliate.
**Figure supplement 4.** Characterizing tubacin action on cilia number in AKAP220KO cells.

suggest that the loss of HDAC6 activity and elevated acetylated tubulin correlate with persistence of primary cilia.

This led to a working hypothesis that AKAP220-targeted HDAC6 modulates primary cilia development. The HDAC6-selective inhibitor tubacin was used as a tool to evaluate the contribution of this enzyme activity in our mIMCD3 cell lines (*Haggarty et al., 2003*; *Figure 3Q*). This drug was applied to test if blocking anchored-HDAC6 activity enhanced ciliary development. In wild-type cells, application of tubacin (2 µM) increased the number of ciliated cells as assessed by immunofluorescent detection of Arl13b (red) and acetylated tubulin (green; *Figure 3R & S*). The gray scale image and quantification of three independent experiments reveal a 3.2-fold increase in percent ciliated cells upon pharmacologically targeting HDAC6 (*Figure 3T & X*). Co-staining with both cilia markers was necessary to delineate between cilia and acetyl-tubulin at the midbodies of dividing cells (*Figure 3S*). Importantly, no change in cilia number occurred when AKAP220-ΔPP1 cells were treated with tubacin (*Figure 3U–Y*). Similarly, AKAP220KO cells were unresponsive to the drug (*Figure 3—figure supplement 4A–D*). This raises the intriguing possibility that an HDAC6-PP1-AKAP220 signaling axis is the intracellular target for the drug tubacin (*Figure 3Z*).

## AKAP220 signaling impacts cortical actin dynamics

The cytoskeleton maintains cell shape and structure by synchronizing the assembly and disassembly of actin, intermediate and tubulin filaments (*Janke and Magiera, 2020*; *Klymkowsky, 1999*). Covalent modification of these elements is an important facet of cytoskeletal regulation (*Portran et al., 2017*). For example, HDAC6 deacetylates the actin-binding protein cortactin to relocate it from the nucleus to its sites of action (*Ran et al., 2015*). Since HDAC6 activity is impaired in AKAP220KO and AKAP220-ΔPP1 cells, it was important to evaluate if nuclear accumulation of acetylated cortactin was enhanced. Immunofluorescent detection of acetyl-cortactin (green) was more prominent in the nuclei of AKAP220KO and AKAP220-ΔPP1 mIMCD3 cells, as compared to wild-type (*Figure 4A,C & E*). This is emphasized in insets and corresponding surface plots (*Figure 4B,D & F*). Amalgamated data from three experiments are presented (*Figure 4G*). Next, we investigated if AKAP220 signaling influences actin filament morphology (*Figure 4H–M*). Immunofluorescent analyses show that AKAP220KO and AKAP220-ΔPP1 cells exhibit dramatic changes in the actin cytoskeleton (red) (*Figure 4H,J & L*). Line plot analyses (40–50 cells) reveal that total loss of the anchoring protein or disruption of HDAC6-PP1 attachment correlates with enhanced accumulation of cortical actin (*Figure 4I,K & M*). These findings pointed towards defects in actin dynamics. Additionally, the distribution of unmodified cortactin was normal in mutant cells, and in a manner consistent with actin.

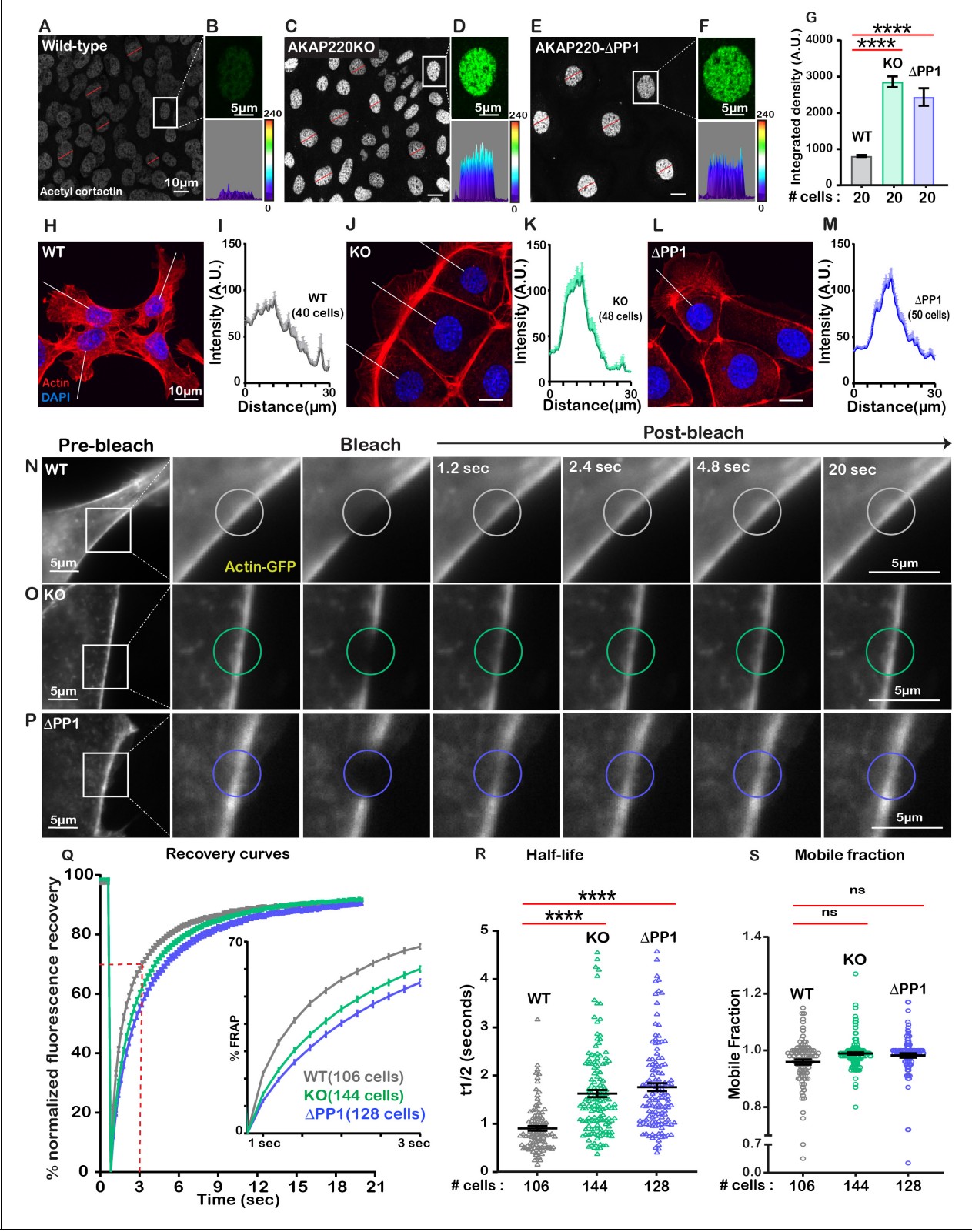

**Figure 4.** Disrupting the AKAP220-PP1 subcomplex impacts actin reorganization. Gray scale images of acetylated cortactin in (**A**) wild-type, (**C**) AKAP220KO, and (**E**) AKAP220-ΔPP1 mIMCD3 cells. Magnified images of nuclei (top) and three-dimensional surface plots (bottom) of acetylated cortactin in (**B**) wild-type, (**D**) AKAP220KO and (**F**) AKAP220-ΔPP1 cells. (**G**) Quantification of amalgamated data (20 cells) from wild-type (gray), AKAP220KO (green), and AKAP220-ΔPP1 (blue) cells. ****p<0.0001, N=3. Scale bars (10 μm). (**H-M**) Confocal images of actin (red) and DAPI (blue) in (**H**)

*Figure 4 continued on next page*

*Figure 4 continued*

wild-type, (J) AKAP220KO, and (L) AKAP220-ΔPP1 cells. Lines indicate sites of line plot analysis to measure actin distribution from nuclei to the lamellipodia in (I) wild-type (gray), (K) AKAP220KO (green) and (M) AKAP220-ΔPP1 (blue) cells. Scale bars (10 μm). (N-S) Fluorescence recovery after photobleaching (FRAP) in mIMCD3 cells. Time course (0–20 s) of GFP-actin imaging in (N) wild-type, (O) AKAP220KO and (P) AKAP220-ΔPP1 cells. (Expanded section) Photobleached portion of cortical actin. (Q) FRAP curves in wild-type (gray), AKAP220KO (green), and AKAP220-ΔPP1 (blue) cells. (Inset) Photo recovery rates over the first 3 s. (R) The t1/2 value for each cell analyzed is presented for wild-type (gray), AKAP220KO (green) and AKAP220-ΔPP1(blue) cells. ****p<0.0001, N=3. (S) The mobile fraction of each cell analyzed is presented for wild-type (gray), AKAP220KO (green), and AKAP220-ΔPP1(blue circles) cells. Non-significant, N=3. Scale bars (5 μm). All error bars are s.e.m. p Values were calculated by unpaired two-tailed Student's t-test. Number of cells analyzed indicated below each column.

The online version of this article includes the following source data and figure supplement(s) for figure 4:

**Source data 1.** FRAP curves for lifeact-GFP in mIMCD3 cells.
**Source data 2.** Half-life measurements for lifeact-GFP in mIMCD3 cells.
**Source data 3.** Mobile fraction measurements for lifeact-GFP in mIMCD3 cells.
**Source data 4.** Intensity of actin in mIMCD3 cells by line plot analysis.
**Figure supplement 1.** Distribution of actin in confluent mIMCD3 cell culture.

Immunofluorescence data depicting actin distribution in confluent cultures of wild-type and AKAP220KO mIMCD3 cell lines is shown in *Figure 4—figure supplement 1A–D*.

To further explore this concept, we performed *Fluorescence Recovery after Photobleaching* (FRAP) at cell-cell junctional actin fibers (*Videos 1–3*). Actin-GFP recovery upon photobleaching was monitored over a time course of 20 s in wild-type (gray; *Figure 4N*); AKAP220KO (green; *Figure 4O*) and AKAP220-ΔPP1(blue; *Figure 4P*) mIMCD3 cells. Recovery curves depict the half-life and mobile fraction (*Figure 4Q*). Expanded section accentuates the expedited recovery of actin in wild-type cells (*Figure 4Q*, inset). The t1/2 for photo-recovery is 1.6 s; n=106 (*Figure 4R*, gray). In contrast, the t1/2 was 0.9 s; n=144 for AKAP220KO cells (*Figure 4R*, green). This is a 0.72 s ±0.09 increase in the rate of photo-recovery when compared to wild-type. Parallel FRAP experiments in AKAP220-ΔPP1 cells calculated a t1/2 of 1.75 s; n=128, representing a 0.85 ± 0.10 s increase in the rate of photo-recovery as compared to wild-type (*Figure 4R*, blue). The availability of actin-GFP was similar in mobile fractions from each cell type (wild-type 95% (gray); AKAP220KO 98% (green); AKAP220-ΔPP1 98% (blue); *Figure 4S*). Thus, disruption of AKAP220-signaling impacts the distribution and dynamics of actin filaments.

## Actin polymerization dictates cilium biogenesis and length

Actin is a key regulator of cilia formation and elongation (*Kim et al., 2015*). AKAP220KO mice exhibit reduced accumulation of apical actin through the diminished GTP-loading of RhoA (*Whiting et al., 2016*). This key regulator of cytoskeletal reorganization contributes to the assembly

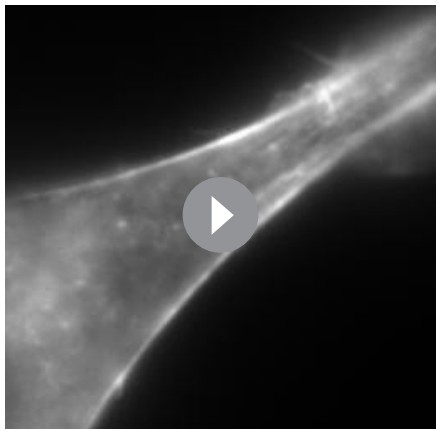

**Video 1.** Recovery of lifeact-GFP upon photobleaching in wild-type mIMCD3 cells.
https://elifesciences.org/articles/67828#video1

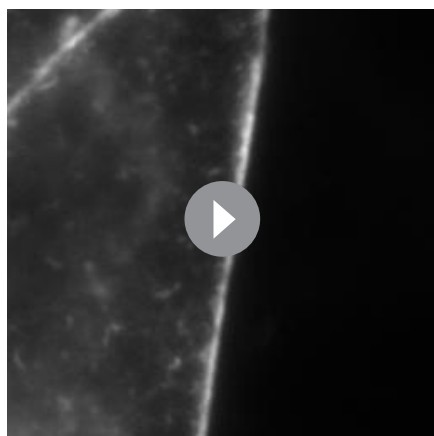

**Video 2.** Recovery of lifeact-GFP upon photobleaching in AKAP220KO mIMCD3 cells.
https://elifesciences.org/articles/67828#video2

of actin barriers in renal cells (*Blattner et al., 2013*). HDAC6-mediated stimulation of actin polymerization is a prerequisite for the disassembly of primary cilia (*Ran et al., 2015*). A convergence of these ideas hypothesizes that defective AKAP220-signaling alters the dynamics of the actin barrier assembly to enhance ciliation (*Figure 5A*).

To test this hypothesis, we treated wild-type cells with the actin-depolymerizing drug Cytochalasin D (*Figure 5B–E*). Drug application (200 nM) for 4 hr favored disassembly of actin (white) as monitored by immunofluorescence (*Figure 5B & D*). Cytochalasin D promoted a 2.95-fold increase in the percentage of ciliated wild-type cells (*Figure 5F*, pink column). AKAP220KO cells displayed longer cilia upon drug treatment (*Figure 5G–L*). These pharmacological effects on cilia length were quantified (*Figure 5M*, pink). Insets feature representative cilia at higher magnification (*Figure 5I & L*, upper panels). Three-dimensional surface plots depict drug-induced changes in cilia length (*Figure 5I &L*, lower panels). Thus, pharmacological blockade of actin assembly augments ciliogenesis. This effect is enhanced in AKAP220 null cells.

Jasplakinolide is a macrocyclic peptide that stabilizes f-actin and augments the formation of actin barriers (*Holzinger, 2009*). Drug treatment (500 nM) for 1.5 hr enhanced detection of cortical actin (white) and decreased the number of ciliated cells (pink; *Figure 5N–S*). Insets feature representative cilia at higher magnification (*Figure 5P & S*, upper panels). Three-dimensional surface plots depict drug-induced changes in cilia length (*Figure 5P & S*, lower panels). Collectively, these pharmacological studies show that bi-directional modulation of actin dynamics directly affects ciliation. This suggests that the actin dynamics changes observed in AKAP220KO and AKAP220-ΔPP1 cells underlie enhanced ciliation.

## AKAP220 signaling influences cilia morphology

We reasoned that a consequence of altered actin dynamics could be changes in cilia morphology. Super-resolution immunofluorescence imaging of fixed cells was performed using Arl13b as a ciliary marker. Cilia appeared retracted in AKAP220-null and ΔPP1-knock in cells in comparison to cylindrical and symmetrical architecture of cilia in wild-type cells (*Figure 6A–D*). A secondary feature was bulbous tips at the distal end of mutant cilia (*Figure 6C & D*). Analyses from three independent experiments show that the occurrence of bulbous tips was prevalent in cilia from AKAP220KO and AKAP220-ΔPP1 cells (*Figure 6E*). To investigate this phenomenon further in living cells, we infected our mIMCD3 cell lines with a lentiviral vector encoding Arl13b-GFP (*Videos 4–6*; *Figure 6F–H*). Quantitative imaging by live-cell super-resolution microscopy revealed that cilia in AKAP220KO (green) and AKAP220-ΔPP1 (blue) cells were 1.6 and 1.7-fold longer than wild-type counterparts (gray; *Figure 6H*). This led us to the conclusion that the bulbous tips presented in *Figure 6C & D* were elongated, flexible cilia that partially retracted (coiled back) on themselves (*Figure 6—figure supplement 1*).

Kidney-on-a-chip technology offers a pseudo-physiological environment that simulates kidney tubules (*Weber et al., 2016*). Microfluidic delivery of nutrients through the lumen of these tubules recapitulates fluid-flow (*Freedman et al., 2013*). This sophisticated tissue-engineering approach was used to evaluate cilia development (*Figure 6I*). Culturing of wild-type mIMCD3 cells formed columnar organoids with few cilia protruding into the lumen (*Figure 6J & K*). In contrast, longer cilia were evident in AKAP220-ΔPP1 organoids (*Figure 6L & M*). Immunofluorescent staining of Arl13b (red) and acetyl tubulin (green) marked cilia and DAPI (blue) detected nuclei. The average cilia length was 3.95-fold greater in AKAP220-ΔPP1 pseudo-tubules (n=73 cilia) as compared to wild-type (n=53 cilia). These effects are more visible in the gray scale images of Arl13b alone (*Figure 6K&M*). Amalgamated data from three independent experiments are presented in *Figure 6N*.

Cilia assembly requires the passage of materials through an actin barrier formed across the basal body (*Farina et al., 2016*). We reasoned that the dynamics of this process may be altered upon manipulation of AKAP220 signaling. Therefore, we combined super-resolution microscopy with *Fluorescence Recovery after Photobleaching* (FRAP) to visualize GFP-Arl13b trafficking into individual cilia (*Figure 6O*). We chose to selectively photobleach the tips of cilia as this approach is less intrusive on the base of cilia (*Blasius et al., 2019*). Arl13b recovery upon photobleaching was monitored over a time course of 2.5 s in wild-type (gray; *Figure 6P*); AKAP220KO (green; *Figure 6Q*) and AKAP220-ΔPP1(blue; *Figure 6R*) mIMCD3 cells. In wild-type cells, the rate of GFP-Arl13b recovery was steady over this time-course. FRAP was more rapid and robust in AKAP220KO and AKAP220-Δ PP1 cilia (*Figure 6S*). Total Arl13b recovered in AKAP220KO is increased 22.35 ± 3% (n = 120 cilia;

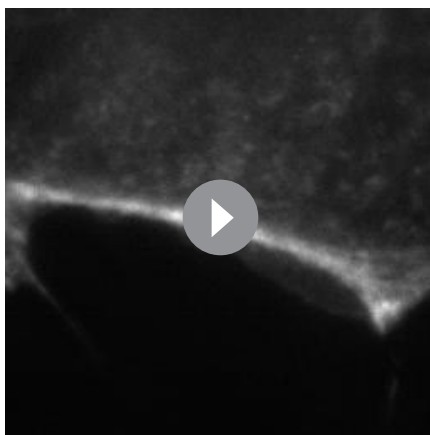

**Video 3.** Recovery of lifeact-GFP upon photobleaching in AKAP220-ΔPP1 mIMCD3 cells.
https://elifesciences.org/articles/67828#video3

green; *Figure 6T*) over wild-type (n=60 cilia; gray; *Figure 6T*). Similarly, recovery of Arl13b in AKAP220-ΔPP1 cilia is 19.04 ± 3% increased (n=126 cilia; blue; *Figure 6T*) over wild-type. The t1/2 of Arl13b recovery was 178.9 ms in wild-type cilia (gray), as compared to 344.7 and 361.6 ms in AKAP220KO and AKAP220-ΔPP1 cilia, respectively (*Figure 6U*; green and blue). These studies suggest that there is a larger mobile pool of Arl13b trafficking into the mutant cilia. One plausible mechanism is that cilia in AKAP220KO and AKAP220-ΔPP1 cells lack an architectural checkpoint that gates protein movement into this organelle. A likely candidate is an f-actin barrier at the base of the cilium (*Figure 6V*).

## HDAC6 inhibition attenuates renal cyst formation

Autosomal dominant polycystic kidney disease (ADPKD) is associated with mutations in *PKD1* and *PKD2,* changes in apical actin and cilia dysfunction (*Halvorson et al., 2010*). This pathology is characterized by fluid-filled cysts that replace normal renal parenchyma (*Figure 7A*). Aberrant HDAC6 activity is a factor in cyst growth and inhibitors of this enzyme are thought to retard cystogenesis (*Cebotaru et al., 2016*). Therefore, we reasoned that pharmacologically targeting HDAC6 with relatively selective inhibitors may reduce cyst formation in cellular models of ADPKD (*Figure 7B*). Human pluripotent stem cells (hPSCs) with a targeted disruption of *PKD2* were differentiated into kidney organoids (*Freedman et al., 2015*). Cysts were identified as large, translucent structures that swayed in response to agitation (*Figure 7C & D*). *PKD2*[-/-] kidney organoids and matched isogenic controls were treated with tubacin (0.2–1 µM) for 48 hr and cyst size was evaluated. At low concentrations of tubacin (0.2 µM), renal cyst size was markedly reduced in *PKD2*[-/-] organoids as compared to the control (*Figure 7E & F*). Similar results were obtained at a higher dose of 1 µM (*Figure 7G & H*). Amalgamated data from five experiments are presented (*Figure 7I*). Drug toxicity as assessed by luminescence assay was evident at higher doses of tubacin (*Figure 7J*). Thus, inhibition of the AKAP220-binding partner HDAC6 impacts the development of cysts in this human kidney organoid model.

## Discussion

The primary cilium is a highly organized mechanosensory transduction unit that responds to environmental cues (*Goetz and Anderson, 2010*; *Wheway et al., 2018*). We have discovered signaling events proceeding through AKAP220 that temper cilia development in kidney collecting ducts. Although it may seem paradoxical that disruption of local signaling can positively impact organellar development, it is important to note that a few AKAPs mitigate signaling events in other cellular contexts (*Bucko and Scott, 2021*; *Langeberg and Scott, 2015*). At neuronal synapses, tonic phosphatase activity constrained by AKAP79/150 attenuates the phosphorylation status and activity of excitatory ionotropic glutamate receptors that contribute to learning and memory (*Hoshi et al., 2005*; *Tunquist et al., 2008*). In cardiomyocytes, AKAP18 anchored phosphodiesterase-3 degrades cAMP that sustains excitation-contraction coupling (*Lygren et al., 2007*). Likewise, AKAP220 binding partners repress aquaporin-2 shuttling to apical membranes of kidney collecting ducts to maintain renal water homeostasis (*Whiting et al., 2016*).

Our imaging analysis of tissue sections from AKAP220KO mice and cell lines indicate that loss of this anchoring protein enhances cilia development. Interestingly, AKAP79/150 has been implicated as a modulator of renal ciliary signaling (*Choi et al., 2011*). However, proximity proteomic approaches identify AKAP220 in cilia, and knockout of the AKAP79/150 ortholog in mIMCD3 cells and mice does not impact cilia development (*May et al., 2020* and *Figure 1—figure supplement 2*). In contrast, our in vivo and in vitro data strongly implicate AKAP220 signaling in the modulation

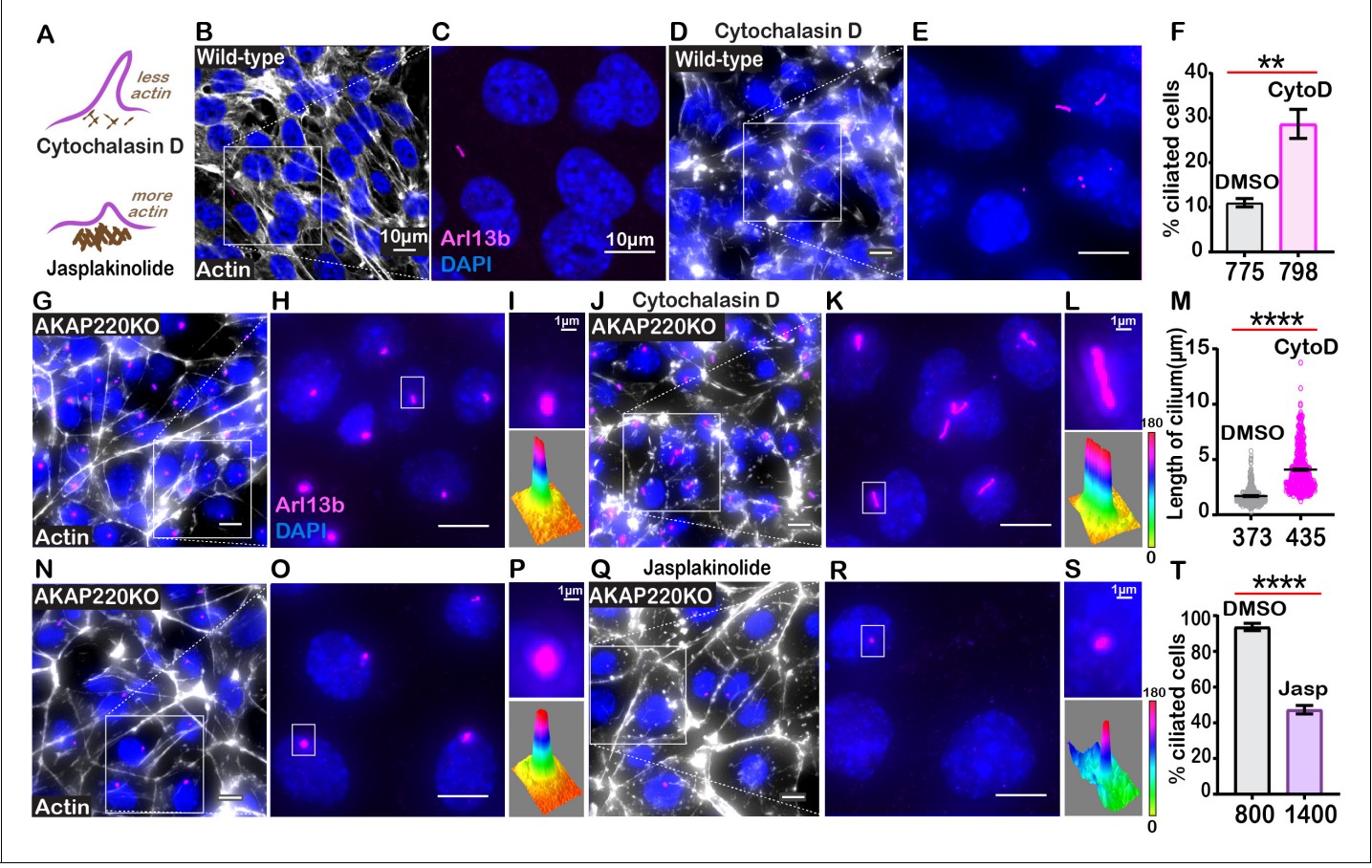

**Figure 5.** Cilia frequency and development involve f-actin assembly. (A) Schematic of how actin modulating drugs impact primary cilia. Cytochalasin D depolymerizes actin barriers. Jasplakinolide stabilizes actin filaments. (B–F) Immunofluorescent detection of actin (white), Arl13b (pink), and DAPI (blue) in wild-type mIMCD3 cells treated with (B) DMSO or (D) 200 nM Cytochalasin D. Enlarged regions emphasize cilia frequency in (C) DMSO and (E) Cytochalasin D-treated cells. (F) Quantification (% ciliated cells) in DMSO (gray) and Cytochalasin D (pink) treated cells. **p<0.01, N=3. (G–M) Immunofluorescent detection of actin (white), Arl13b (pink), and DAPI (blue) in (G) DMSO and (J) Cytochalasin-D-treated AKAP220KO cells. (Inset) Expanded field of cells treated with (H) DMSO or (K) Cytochalasin D. Boxed regions in (I and L) focus on a single cilium (top) and three-dimensional surface plot (bottom). The width of the cylindrical region in the 3D surface plot represents cilium length. (M) Quantification of cilia length in DMSO (gray) and Cytochalasin D (pink) treated cells. ****p<0.0001, N=3. (N–T) Immunofluorescent staining of actin (white) and DAPI (blue) of (N) (DMSO) and (Q) (Jasplakinolide)-treated AKAP220KO cells. (Inset) Expanded field of cells treated with (O) DMSO or (R) Jasplakinolide. Boxed regions in (P and S) focus on a single cilium (top) and three-dimensional surface plot (bottom). The width of the cylindrical region in the 3D surface plot represents cilium length. (T) Quantification (% ciliated cells) in DMSO (gray) and Jasplakinolide (purple). ****p<0.0001, N=3. All error bars are s.e.m. p Values were calculated by unpaired two-tailed Student's t-test. Scale bars (10 µm). Number of cells analyzed indicated below each column.

The online version of this article includes the following source data and figure supplement(s) for figure 5:

**Source data 1.** Percent ciliated in DMSO or Cytochalasin-D-treated wild-type cells.

**Source data 2.** Cilia length in in DMSO or Cytochalasin-D-treated AKAP220KO cells.

**Source data 3.** Percent ciliated in DMSO or Jasplakinolide-treated AKAP220KO cells.

**Figure supplement 1.** Characterizing the effect of actin-modulating drugs on AKAP220-ΔPP1 cilia.

of aquaporin-2 trafficking and ciliogenesis. This could occur through one of two mechanisms- either the cellular signals that attenuate aquaporin-2 trafficking and diminish cilia development are processed through the same macromolecular complex, or, distinct signaling islands of AKAP220-binding partners are assembled to control each process.

A unifying principle of our study is that AKAP220-binding partners affect the development of primary cilia by enacting cytoskeletal changes at the level of actin polymerization and tubulin acetylation. This implicates deacetylation as a fundamental signal termination process that checks the rate of cilia development. Histone deacetylase 6 (HDAC6) that targets the cytoskeletal elements alpha

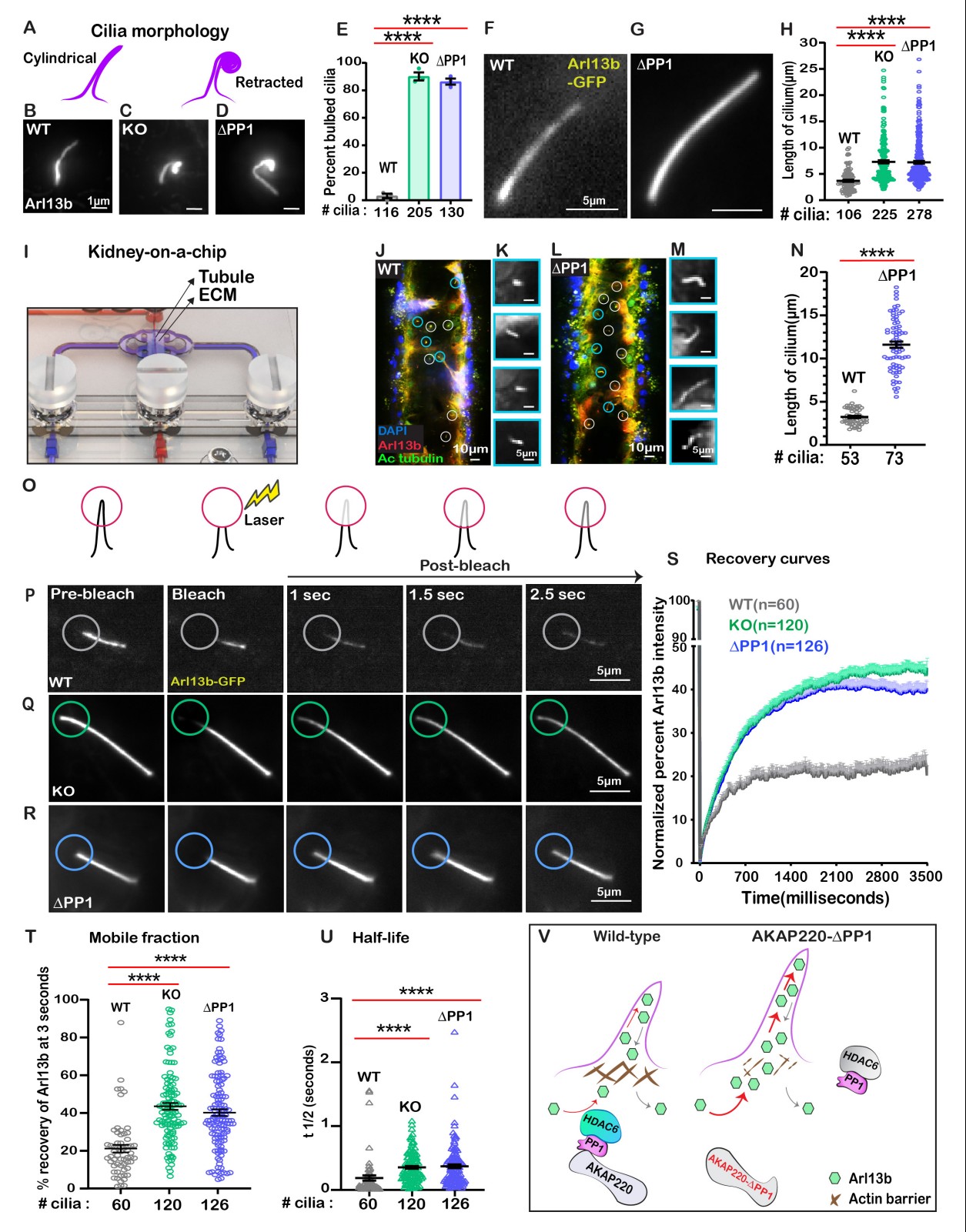

**Figure 6.** Loss of phosphatase anchoring promotes cilium elongation. (**A**) Schematic of cylindrical and retracted cilia morphologies (purple). Super-resolution fixed-cell imaging of Arl13b (gray scale) in (**B**) wild-type, (**C**) AKAP220KO, and (**D**) AKAP220-ΔPP1 cilia. (**E**) Quantification (% bulbed cilia) in wild-type (gray), AKAP220KO (green), and AKAP220-ΔPP1 (blue) cells. ****p<0.0001, N=3. Scale bars (1 μm). Super-resolution live-cell images of Arl13b-GFP in (**F**) wild-type and (**G**) AKAP220-ΔPP1 cilia. (**H**) Quantification of cilia length in wild-type (gray), AKAP220KO (green), and AKAP220-ΔPP1 (blue)

*Figure 6 continued on next page*

*Figure 6 continued*

cells. ****p<0.0001, N=3. Scale bars (5 μm). (I) Kidney-on-a-chip device. Location of kidney tubule (Tubule) and extracellular matrix (ECM) are indicated. Confocal imaging of (J) Wild-type and (L) AKAP220-ΔPP1cells cultured in chip device with Arl13b (red), acetyl tubulin (green), and DAPI (blue). Cilia (white circles) are marked. (Insets) Magnified images of representative cilia (cyan circles) from (K) wild-type and (M) AKAP220-ΔPP1 pseudotubules. (N) Quantification of cilia length in wild-type (gray) and AKAP220-ΔPP1 (blue). ****p<0.0001, N=3. Scale bars (10 μm). Inset scale bars (5 μm). (O–U) Fluorescence recovery after photobleaching (FRAP) of Arl13b-GFP in primary cilia. (O) Schematic of how FRAP was measured. (P–R) Super-resolution live-cell images of Arl13b-GFP in (P) wild-type, (Q) AKAP220KO and (R) AKAP220-ΔPP1 cells. Circles mark bleached portion of cilia. (S) FRAP recovery curves of Arl13b over time (3500 ms) in wild-type (gray), AKAP220KO (green), and AKAP220-ΔPP1 (blue) cells. Scale bars (5 μm). (T) Quantification of mobile fraction in wild-type (gray), AKAP220KO (green), and AKAP220-ΔPP1 (blue) cilia. ****p<0.0001, N=3. (U) Half-life of Arl13b in wild-type (gray), AKAP220KO (green), and AKAP220-ΔPP1 (blue) cilia. ****p<0.0001, N=3. (V) Schematic depicting how AKAP220 modulation of the actin barrier influences movement of proteins in and out of primary cilia. All error bars are s.e.m. p Values were calculated by unpaired two-tailed Student's t-test. Number of cilia analyzed indicated below each column.

The online version of this article includes the following source data and figure supplement(s) for figure 6:

**Source data 1.** Primary cilia length from live cell experiments in mIMCD3 cells.
**Source data 2.** Length of primary cilia in kidney on a chip device.
**Source data 3.** FRAP curved for Arl13b-GFP in mIMCD3 cell lines.
**Source data 4.** Mobile fraction of Arl13b-GFP 3 s after photobleaching.
**Source data 5.** Half-life of Arl13b-GFP in mIMCD3 cell lines.
**Figure supplement 1.** Super resolution videos depicting flexibility of AKAP220KO cilia.

---

tubulin and cortactin has been identified as a driver of cilia disassembly (*Ran et al., 2015*). In concordance with this notion, data in *Figure 2V* show that rescue upon overexpression of HDAC6 in an AKAP220KO background restores control of ciliation. Several lines of evidence suggest a role for AKAP220 in this process. Data in *Figure 2* indicate that this deacetylase is more labile in the absence of the anchoring protein. While phosphorylation protects HDAC6 from ubiquitination, and dephosphorylation favors proteasomal degradation, it has been unclear how this enzyme is maintained in proximity of developing cilia (*Ran et al., 2020*). Our findings point towards a previously unrecognized adaptor function for protein phosphatase 1 (PP1). This reasoning is predicated on evidence that AKAP220 is a conventional PP1-targeting subunit that utilizes a KVxF motif to contact the phosphatase (*Bollen et al., 2010*).

Although the mechanism of HDAC6 interaction with PP1 is less clear (*Brush et al., 2004*), it appears that the anchored phosphatase retains the capacity to function as an adaptor protein and recruit this additional binding partner. Several lines of evidence prompted us to investigate this further- pulse-chase data showing that HDAC6 protein stability is compromised in AKAP220KO cells and proximity-ligation data detecting HDAC6 in AKAP220-signaling islands presented in *Figure 2I–O*. Further, co-immunoprecipitation experiments presented in *Figure 3D* argue that removal of the principle PP1-interaction motif on AKAP220 negatively impacts association with HDAC6. Hence, loss of PP1-anchoring uncouples HDAC6 association with AKAP220 signaling islands.

However, our findings are equally consistent with the notion that removal of the KVXF motif on AKAP220 impedes, but does not fully abolish, PP1-anchoring. As mentioned earlier, we previously identified three secondary PP1 interactive surfaces on AKAP220 (*Schillace and Scott, 1999*; *Schillace et al., 2001*). Subsequently, the notion of multisite contact between serine/threonine phosphatases and their binding partners has become a standard view (*Hoermann et al., 2020*). Since it is not technically feasible to remove all four PP1 interactive surfaces on AKAP220 and retain a properly folded protein, we can only conclude that there is a reduced association of the PP1-HDAC6 sub complex in the context of cells expressing AKAP220-ΔPP1. Two plausible mechanistic explanations that support this conclusion. First, removal of the primary PP1 targeting motif on AKAP220 lessens binding of the PP1-HDAC6 subcomplex. Second, mutation of the KVXF motif could expose secondary HDAC6 binding determinants that could reside within AKAP220 or any of its other binding partners. Irrespective of either mechanism, we have definitively shown

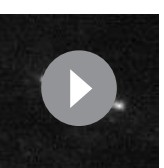

**Video 4.** Recovery of Arl13b-GFP upon photobleaching in wild-type mIMCD3 cells.
https://elifesciences.org/articles/67828#video4

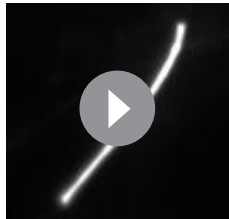

**Video 5.** Recovery of Arl13b-GFP upon photobleaching in AKAP220KO mIMCD3 cells.
https://elifesciences.org/articles/67828#video5

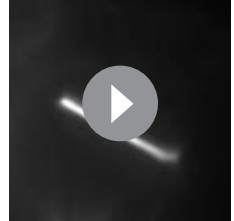

**Video 6.** Recovery of Arl13b-GFP upon photobleaching in AKAP220-ΔPP1 mIMCD3 cells.
https://elifesciences.org/articles/67828#video6

that HDAC6 interacts with AKAP220. Moreover, loss of PP1 anchoring significantly impacts maintenance of HDAC6 within AKAP220 signaling islands. Thus, AKAP220-PP1 subcomplexes create a platform for the targeting of HDAC6 to repress tubulin acetylation during ciliogenesis. This may represent a homeostatic mechanism that enables cells to enter mitosis.

Mutations in the ciliary phosphoproteins polycystin 1 and 2 are linked to autosomal dominant polycystic kidney disease (ADPKD) (*Hughes et al., 1995*; *Mochizuki et al., 1996*; *Streets and Ong, 2020*). Trafficking of polycystin-1 to the ciliary membrane is dependent on local PP1 activity (*Parnell et al., 2012*). Likewise, bi-directional phosphorylation-dependent control of polycystin-2 channel conductance is modulated by an anchored PKA-PP1 component (*Streets et al., 2013*). Thus, local dephosphorylation events are key to the termination of mechanotransduction signals that govern primary cilia action in ADPKD. Using the AKAP220-ΔPP1 knock-in mutant as a mechanistic probe, we have uncovered a new non-catalytic role for anchored-PP1 in the regulation of cytoskeletal events that underlie renal cilia biogenesis. The striking changes observed in actin remodeling and cilia morphology presented in *Figures 4* and *5* can solely be attributed to disruption of the KVXF

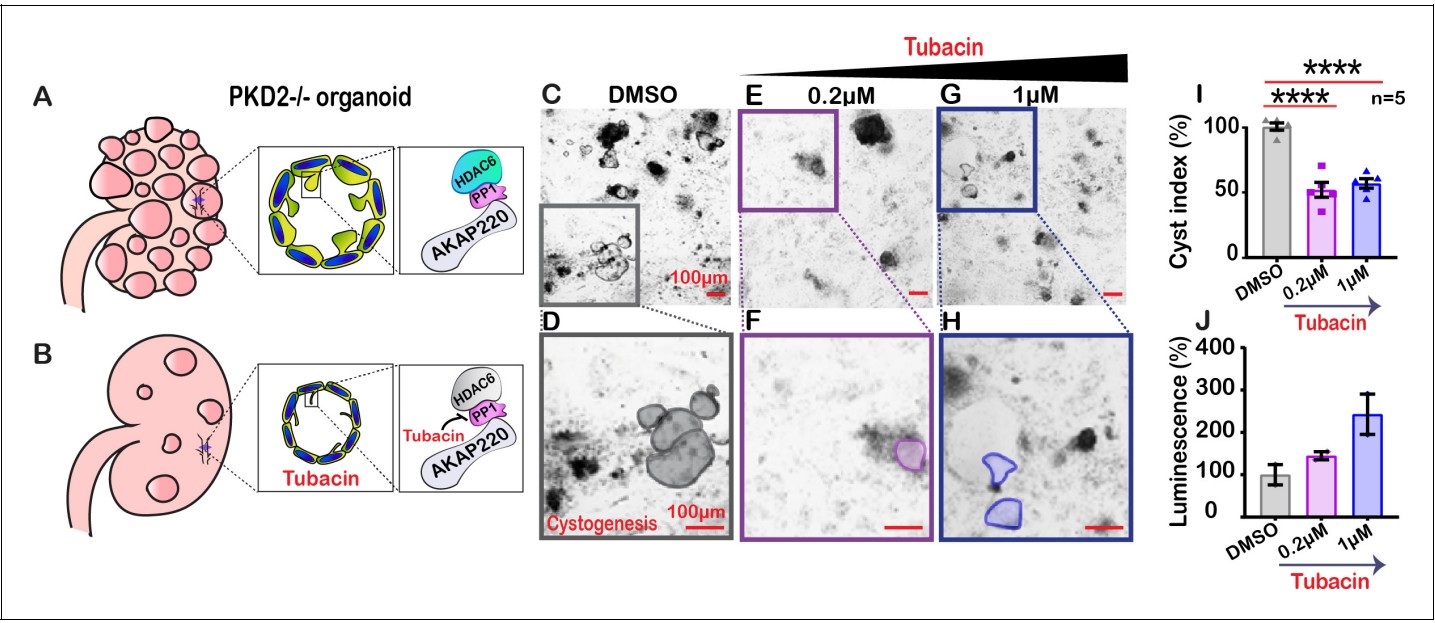

**Figure 7.** Inactivating HDAC6 reduces renal cystogenesis. Schematic of polycystic kidneys (**A**) before and (**B**) after tubacin treatment. (Insets) Tubacin action on the AKAP220-signaling complex in precystic PKD2$^{-/-}$ organoids derived from WA-09 cells. (**C-J**) Confocal images of PKD2$^{-/-}$ organoids treated with (**C**) DMSO, (**E**) 0.2 μM and (**G**) 1 μM tubacin. Enlarged regions from (**C, E and G**) showing cysts in (**D**) DMSO, (**F**) 0.2 μM and (**H**) 1 μM tubacin-treated conditions. Scale bars (100 μm). (**I**) Quantification (% cyst index) in DMSO (gray), 0.2 μM (purple) and 1 μM (blue) tubacin-treated conditions. ****p<0.0001, N=5. (**J**) Luminescence assay to detect toxicity of the drug is plotted for DMSO (gray), 0.2 μM (purple) and 1 μM (blue) tubacin-treated conditions. Error bars are s.e.m. p Values were calculated by unpaired two-tailed Student's t-test.

The online version of this article includes the following source data for figure 7:

**Source data 1.** Quantification of cyst size in DMSO and tubacin-treated iPSCs.

motif in AKAP220 (*Schillace and Scott, 1999*). Concomitant effects on HDAC6 location not only enhance cortactin acetylation, but also enact changes in actin dynamics. An additional factor to consider is that cortactin may also contribute to ciliogenesis via mechanisms that are complementary to signals proceeding through AKAP220 (*Bershteyn et al., 2010*).

Such cytoskeletal reorganization at defined sites is a prelude to cilia extension. Data in *Figure 6F–N* indicate that these effects appear to be more pronounced in live cell experiments and within the pseudo physiological environment of the kidney-on-a-chip. Focal adhesion proteins attach the basal body to actin filaments (*Drummond et al., 2018*; *Kim et al., 2015*). This initiates clearing of the local actin barrier to commence microtubule nucleation. In keeping with this notion, data in *Figure 5* show that the actin depolymerizing drug Cytochalasin D enhances cilia persistence and length, whereas the actin-stabilizing compound Jasplakinolide has the opposite effect (*Casella et al., 1981*; *Holzinger, 2009*). These pharmacological tools highlight the action of AKAP220-associated enzymes at the level of the actin barrier. This substructure is part of a 'ciliary necklace' that acts as a physical checkpoint for proteins moving into the cilium (*Long and Huang, 2019*). This is supported by photobleaching studies showing that Arl13b moves more readily in AKAP220-ΔPP1 mutant cilia and these organelles are approximately two fold longer than wild-type (*Figure 6H*). Thus, it is plausible that AKAP220-associated PP1 participates in maintenance of an actin barrier close to the basal body of the cilium, the loss of which leads to unhindered protein movement into the developing organelle.

Although lesions in cilia and polycystins are linked to Autosomal Dominant Polycystic Kidney Disease (ADPKD), the molecular details remain unclear and are inherently paradoxical (*Hughes et al., 1995*; *Mochizuki et al., 1996*). Thus, searching for therapeutic approaches that modify local protein acetylation to restore functional primary cilia is a rationale behind our concluding studies in *Figure 7*. There are several important caveats to consider. How primary cilia can be pro-cystogenic in one context yet anti-cystogenic in another context remains a conundrum in the field (*Kolb and Nauli, 2008*; *Lin et al., 2003*; *Ma et al., 2013*). This apparent 'cilia paradox' limits our ability to accurately predict how pharmacological agents targeting cilia might affect cellular models of cystogenesis or more sophisticated in vivo models. Nonetheless, HDAC6 inhibitors have been used by other groups to target cyst size (*Cebotaru et al., 2016*; *Yanda et al., 2017*). Thus, mechanisms by which tubacin may act to inhibit cystogenesis could be less than straightforward. Particularly, given the multitude of functions for HDAC6 including control of cell morphology, cell adhesion, cell migration and tumor metastasis. Interestingly, these are known sites of action of the AKAP220 signaling complexes (*Boyault et al., 2007*; *Logue et al., 2011*; *Wong and Scott, 2004*). Thus, it is reasonable to speculate that tubacin could target the AKAP220-PP1-HDAC6 axis in these cellular contexts. On the other hand, HDAC6 also modulates a variety of cellular processes independently of the anchoring protein. Thus, we propose that tubacin action at multiple sites including the AKAP220-PP1-HDAC6 axis could impact the anti-proliferative effect on cyst formation. Therefore, pharmacologically targeting anchored-HDAC6 may not only enhance cilia development, but also interfere with certain downstream signaling events that contribute to cystogenesis. Finally, the AKAP-targeting concept has recently been used to restrain kinase inhibitor drugs at defined subcellular locations (*Bucko et al., 2019*; *Bucko et al., 2020*). We intend to expand this approach toward developing a precision pharmacology strategy to selectively deliver HDAC6 inhibitors to the base of primary cilia for the treatment of ciliopathies including chronic polycystic kidney diseases.

## Materials and methods

**Key resources table**

| Reagent type (species) or resource | Designation | Source or reference | Identifiers | Additional information |
|---|---|---|---|---|
| Antibody | Acetylated tubulin | Thermo Fisher | Mouse monoclonal 32–2700 RRID:AB_2533073 | IF(1:5000), WB(1:1000) |
| Antibody | ActinRed-555 | Molecular probes | R37112 | Manufacturer instructions |

*Continued on next page*

*Continued*

| Reagent type (species) or resource | Designation | Source or reference | Identifiers | Additional information |
|---|---|---|---|---|
| Antibody | Arl13b | Proteintech | Rabbit Polyclonal 17711–1-AP RRID:AB_2060867 | IF(1:5000) |
| Antibody | AKAP220 | Santa Cruz Biotech | Goat polyclonal No longer available | IF (1:500) |
| Antibody | AKAP220 | Rockland | Rabbit polyclonal Custom generated | WB (1:500) |
| Antibody | Amersham ECL Mouse IgG, HRP-linked F(ab')$_2$ fragment (from sheep) | GE Life Sciences | NA9310 RRID:AB_772193 | WB (1:10000) |
| Antibody | Amersham ECL Rabbit IgG, HRP-linked F(ab')$_2$ fragment (from donkey) | GE Life Sciences | NA9340 RRID:AB_772191 | WB (1:10000) |
| Antibody | Donkey anti-goat IgG-HRP | Santa Cruz | sc-2020 RRID:AB_631728 | WB (1:10000) |
| Antibody | Donkey anti-Mouse IgG, Alexa Fluor 488 | Invitrogen | A-21202 RRID:AB_141607 | IF (1:500) |
| Antibody | Donkey anti-Mouse IgG, Alexa Fluor 555 | Invitrogen | A-31570 RRID:AB_2536180 | IF (1:500) |
| Antibody | Donkey anti-Mouse IgG, Alexa Fluor 647 | Invitrogen | A-11126 RRID:AB_221538 | IF (1:500) |
| Antibody | Donkey anti-Rabbit IgG, Alexa Fluor 488 | Invitrogen | A-21206 RRID:AB_2535792 | IF (1:500) |
| Antibody | Donkey anti-Rabbit IgG, Alexa Fluor 555 | Invitrogen | A-31572 RRID:AB_162543 | IF (1:500) |
| Antibody | Donkey anti-Rabbit IgG, Alexa Fluor 647 | Invitrogen | A-31573 RRID:AB_2536183 | IF (1:500) |
| Antibody | GFP | Rockland | 600-101-215 goat polyclonal RRID:AB_218182 | WB (1:1000), IF (1:500) |
| Antibody | HDAC6 | Proteintech | 12834–1-AP Rabbit Polyclonal RRID:AB_10597094 | IF (1:200) |
| Antibody | V5 | Invitrogen | R960-25 Mouse monoclonal RRID:AB_2556564 | IF (1:500) |
| Cell line (*H. sapein*) | HEK293 | | RRID:CVCL_0045 | Maintained in Scott lab in DMEM supplemented with 10% FBS |
| Cell line (*M. musculus*) | mIMCD3 | | RRID:CVCL_0429 | Maintained in Scott lab in DMEM/F12 supplemented with 10% FBS |
| Chemical compound, drug | DAPI | Thermo Fisher | 62248 RRID:AB_2307445 | IF (1:1000) |
| Chemical compound, drug | Dimethylsulfoxide (DMSO) | Pierce | TS-20688 | Manufacturer's instructions |
| Chemical compound, drug | DMEM FluoroBrite | Life Technologies | A1896701 | |
| Chemical compound, drug | DMEM/F-12 Hepes | Life Technologies | 11330057 | |
| Chemical compound, drug | DMEM, high glucose | Life Technologies | 11965118 | |
| Chemical compound, drug | Fetal Bovine Serum | Thermo Fisher | A3382001 | |

*Continued*

| Reagent type (species) or resource | Designation | Source or reference | Identifiers | Additional information |
|---|---|---|---|---|
| Chemical compound, drug | Lipofectamine 2000 Transfection Reagent | Invitrogen | 11668027 | |
| Chemical compound, drug | NuPAGE LDS Sample Buffer 4X | Thermo Fisher | NP0008 | |
| Chemical compound, drug | Opti-MEM, Reduced Serum Medium, no phenol red | Life Technologies | 11058021 | |
| Chemical compound, drug | ProLong Diamond Antifade Mountant | Life Technologies | P36961 RRID:SCR_015961 | Manufacturer's instructions |
| Chemical compound, drug | Polybrene | Santa Cruz | 134220 | Manufacturer's instructions |
| Chemical compound, drug | Super Signal West Dura Extended Duration Substrate | Thermo Fisher | 34075 | |
| Chemical compound, drug | TransIT-LT1 Transfection Reagent | Mirus | MIR2300 | |
| Chemical compound, drug | Trypsin-EDTA (0.25%), phenol red | Gibco | 25200056 | |
| Commercial assay or kit | BCA Protein Assay Kit | Thermo Fisher | 23227 | |
| Recombinant DNA reagent | pMD2.G | | RRID:Addgene_12259 | gift from Didier Trono; Addgene plasmid #12259 |
| Recombinant DNA reagent | psPAX2 | | RRID:Addgene_12260 | gift from Didier Trono; Addgene plasmid #12259 |
| Recombinant DNA reagent | Arl13b-GFP | | RRID:Addgene_40879 | gift from Tamara Caspary; Addgene plasmid # 40879 |
| Recombinant DNA reagent | lifeact-GFP | | RRID:Addgene_58470 | gift from Dyche Mullins; Addgene plasmid # 58470 |
| Recombinant DNA reagent | pcDNA-HDAC6-flag | | RRID:Addgene_30482 | gift from Tso-Pang Yao; Addgene plasmid # 30482 |
| Recombinant DNA reagent | pEGFP-HDAC6-DC | | RRID:Addgene_36189 | gift from Tso-Pang Yao; Addgene plasmid # 36189 |
| Software, algorithm | Fiji/ImageJ | ImageJ (http://imagej.nih.gov/ij/) | | |
| Software, algorithm | GraphPad Prism | GraphPad Prism (https://graphpad.com) | | |
| Software, algorithm | SoftWoRx | GE Healthcare | | |
| Other | Bolt 4–12% Bis-Tris Plus Gels | Invitrogen | NW04120BOX | |

## Generation of AKAP220KO mice

### Knockout vector

The KO vector was generated using a Bacmid (Children's Hospital Oakland Research Institute, clone RP23-239B21). Vega Biolabs engineered a retrieval plasmid containing arms of homology flanking the target sequence to capture ~14 kb of genomic *Akap11* locus. LoxP sites were added to intronic regions surrounding exons 6 and 7 and a Neo cassette flanked by two Frt sites (removed by breeding with mice expressing FLP recombinase). *HSV-TK*I cassette served as negative selection marker (outside the *Akap11* genomic sequence). ES cell clones were grown in the presence of fialuridine. Southern blot band shift identified incorporation of BamHI and StuI sites into genomic DNA. Further details of ES cell electroporation, Cell screening, chimera production and FRT-FLP and Cre-lox breeding of AKAP220KO mice can be found in our earlier publication (*Whiting et al., 2016*).

### Genotyping

Genotyping the floxed *Akap11* allele was done with primers used in ES cell screening. A second PCR primer pair annealing outside the loxP sites detected loxP collapse in $Akap11^{-/-}$ mice. Full genotyping of global KO animals requires the WT and collapsed PCR reactions. Heterozygous animals have PCR products present in both reactions, whereas homozygotes are positive for only the WT or collapsed allele.

## Tissue section immunofluorescent staining

Wild-type and AKAP220KO mice kidneys were fixed in 10% (vol/vol) buffered formalin (4°C), embedded in paraffin and 4-μm-thick sections collected. Sections were deparaffinized using Citrasolv (Fisher) and antigen retrieved in buffer A using a Retriever 2100 pressure cooker (Electron Microscopy Sciences). Tissue sections were blocked in 10% (vol/vol) donkey serum in PBS solution before overnight incubation with the respective primary antibodies.

Cell culture mIMCD3 cells were maintained in DMEM: F12 1:1 media supplemented with 10% FBS and penicillin/streptomycin. Cells were transfected with plasmids using Mirus TransIT-LT1 transfection reagent and incubated for 48–72 hr before lysis or fixation. All cell lines were maintained in a 5% CO2 incubator at 37°C.

mIMCD3 spheroids mIMCD3 cells were seeded in Matrigel to generate spheroids as previously described (*Giles et al., 2014*). The spheroids were stained with Acetyl tubulin for marking primary cilia and those with a visible open lumen were imaged and used in the quantification.

## Generation of AKAP220-ΔPP1, AKAP150KO, and AKAP220-150DKO cells by CRISPR-Cas9

A 20-nucleotide sequence followed by a protospacer adjacent motif from *Streptococcus pyogenes* targeting the second exon of mouse *Akap11* was selected. The guide sequence was selected for the least number of potential off-target sites. The guide for AKAP220-ΔPP1 (protospacer adjacent motif CGG) was as follows: 5′- GCAAACTGGACTTTTTTCCC-3′. The plasmid was transfected into mIMDC3 cells by Lipofectamine 2000. Two days after transfection, single cells were sorted into 96-well plates by FACS. Two weeks after sorting, confluent clones were analyzed by genomic PCR. PCR products were sequenced to identify the mutation TATA (5′ ACTGCTACTGCA 3′) in place of KVQF (5′ AAAGTCCAGTTT 3′) in the genomic sequence.

The AKAP150KO mIMCD3 cells were made using the Crispr-Cas9 plasmid purchased from Santa Cruz biotech (sc-433620) and transfection and selection was performed as described above. The AKAP220-150 DKO mIMCD3 cells were generated using the same guides as AKAP150KO in an AKAP220KO background.

### Virus generation

Constitutively active lentiviral plasmids expressing Arl13b-GFP were transfected into HEK cells along with viral packaging and envelope plasmids. The viral particles generated are filtered and added to mIMCD3 cells. After 24 hr of incubation, the cells were selected in zeocin (400 ug/ml) for the next 2 weeks. After selection, the cells were trypsinized and plated at a low density of about 1 cell/well in 96-well plate, expanded and tested for expression of Arl13b-GFP by western blotting and immunofluorescence.

### Immunoblotting and blot analysis

Cells were grown to the desired confluence and washed once with PBS at room temperature. Cold lysis buffer (20 mM HEPES, [pH 7.4], 150 mM NaCl, 1 mM EDTA, 1% triton X-100 in water) was added along with protease and phosphatase inhibitors and the plate was rocked gently at 4°C for 10 min. The cell lysate was then scarped into a pre-chilled tube and cleared at 12,000*g for 10 min at 4°C. A BCA assay (Pierce) was used to determine protein concentrations, and 30 μg of protein was loaded onto a Bolt 4–12% bis-Tris gel (Life Technologies). The cleared lysate was boiled in 2X SDS loading buffer for 10 min before loading. Proteins were transferred to nitrocellulose membrane and blocked in either 5% milk. The blot was incubated in primary antibody at 1:1000 or as specified by the manufacturer overnight at 4°C. Immunoblots were washed (three times, 10 min each) in TBST before incubation in a 1:10,000 secondary antibody for 1 hr at room temperature. Immunoblots

were washed again in TBST (three times, 10 min each) before imaging on an iBright FL1000 (Thermo Fisher Scientific) with SuperSignal Dura ECL reagent (Thermo Fisher Scientific). Densitometry for blot quantification was done using thermo fisher's software.

## Sample preparation

Cells were seeded on acid-washed coverslips in 12-well plates. After they achieve the desired confluence, the wells were rinsed thrice with PBS and fixed with 4% paraformaldehyde in PBS or 10% ice cold methanol (based on the antibody specification) for 12 mins. After fixation, the cells were permeabilized in PBS+0.1%Triton x-100+1%BSA for 12 min, blocked with 2%BSA for 30 min and treated with the respective primary antibodies overnight. After thorough washing in PBS the next day (three times, 5 min each), secondary antibodies conjugated to Alexa fluor dyes were added for 2 hr. After two washes in PBS, DAPI along with or without the actin probe (based on the experiment) was added for 10 min. The coverslips were washed two more times in PBS and mounted on slides using ProLong Diamond Antifade Mountant (Life Technologies).

## Imaging and analysis

Cells were imaged on a Keyence BZ-X710 microscope (Keyence, Itasca, IL) using the relevant filter cubes for DAPI (blue filter), Actin (red filter), Cortactin (green filter). All images were acquired with the same magnification (100X, oil immersion), exposure time, and illumination intensity. Images were quantified and processed using ImageJ software.

## Proximity ligation assay (PLA)

HEK293T cells were grown on coverslips and transfected with V5-tagged AKAP220 and EGFP. Transfections were performed according to manufacturer's protocol for Mirus-LT1 reagent. Twenty-four hrs after transfection, cells were fixed with 4% paraformaldehyde/PBS for 15 min. Cells were then subjected to PLA according to manufacturer's instructions (Duolink Proximity Ligation Assay, Sigma). V5 (Invitrogen, #R960-25, 1:500) and HDAC6 (Proteintech, #12834–1-AP, 1:200) antibodies were used to probe the samples. Z-stacks of fluorescent images were collected using a Keyence BZ-X710 with relevant filter cubes. Maximum intensity projections were made for each image and regions of interest (ROIs) were selected based on EGFP expression. Puncta number and fluorescence intensity were measured by automation using Keyence hybrid cell counter set to detect thresholded puncta between 0 and 1.0 µm in diameter. Total cell number per ROI was determined by counting DAPI-stained nuclei.

## FRAP (Fluorescence recovery after photobleaching)

### Sample preparation

Cells were reverse transfected with lifeact-GFP (Addgene plasmid no.58470) and seeded into 60 mm glass-bottom plates for 24 hr. The transfected plates are rinsed in PBS and incubated in Fluorobrite medium containing NucBlue Hoescht 33342 stain (R37605, Invitrogen, one drop/ml). For Arl13b FRAP, mIMCD3 lines were transduced with L13-Arl13bGFP lentiviral construct and selected as outlined above. These cells were seeded in 60 mm glass-bottom plates. They were treated with 0.1%FBS for 24 hr and then the experiment was performed.

### Imaging technique and analysis

Cells were imaged on a GE Deltavision OMX SR microscope (GE Life Healthcare Sciences). After loading (as above), cells were placed in a humidified chamber with 5% CO2 at 37°C and imaged using a 60X oil immersion objective (Olympus, Shinjuku, Tokyo, Japan). The actin signal was photobleached in the green channel (488 nm) using a 15% laser power for 0.05 s. A total of five events were captured before photobleaching. Images after photobleaching were captured every 100 ms for 10 s. For Arl13b FRAP, Lentiviral Arl13b GFP mIMCD3 cells. The Laser pulse at 488 was at 30% T, in a spot at the tip of the cilium to bleach it for 0.1 s. Five events before the bleach and a total of 350 time points were taken at 10msec increments. All images were acquired with the same settings. Images were quantified and processed using ImageJ software.

## Antibodies

The following antibodies were used in this study for immunoblotting: AKAP220 (custom rabbit polyclonal,1:1000), GAPDH (Novus, mouse monoclonal, 1:5000), Acetylated tubulin (Thermo Fisher # 32–2700, 1:1000), HDAC6 (Proteintech # 12834–1-AP, 1:1000). Antibodies used for immunofluorescence: Acetylated tubulin (Thermo Fisher # 32–2700, 1:10000), HDAC6 (Proteintech, 12834–1-AP, 1:400), Arl13b (Proteintech # 17711–1-AP, 1:5000), Actin (Molecular probe, R37112, 1drop/ml).

## Plasmids

Lentiviral Arl13b-GFP (# 40879), pcDNA-HDAC6-flag (#30482), pEGFP-HDAC6-DC (#36189), and lifeact-GFP (#58470) were purchased from addgene. V5-AKAP220 was generated in the lab.

## Statistics for all experiments

Statistical analysis was performed using an unpaired two-tailed Student's t-test or one-way ANOVA (based on the number of samples) in GraphPad Prism software. All values are reported as mean ± standard error of the mean (s.e.m) with p-values less than 0.05 considered statistically significant. For each experiment, number of independent experiments (N) and number of individual points from all experiments (n) are presented.

## Line-plot analysis in ImageJ

For the actin and cortactin distribution experiments, lines were drawn from the outer edge of the nucleus to beyond the lamellipodia of the cell and line plots were generated using ImageJ to give the pixel intensity of each of the proteins along the line. The integrated density values were plotted in Prism to generate a graphical representation of protein distribution in each cell type.

## HDAC6 activity assay

The experiment was performed using BioVision's HDAC6 activity assay kit. mIMCD3 cells were seeded in 10 cm dishes and allowed to grow for the desired confluence. They were then lysed using the lysis buffer provided in the kit. The protein concentration was measured using BCA assay kit (Pierce) and 20 µg of protein was loaded per well. We used 5µM tubacin in the experiment and measured the fluorescence intensity at 380/490 nm in a plate reader.

Drug treatment experiments mIMCD3 cells were seeded on cover slips in 12-well plates and allowed to grow to the desired confluence. They were treated with 2µM tubacin for 4 hr, 200 nM cytochalasin D for 4 hr or 500 nM Jasplakinolide for 1.5 hr and fixed immediately after treatment (as previously described). The cells were stained for acetyl tubulin and Arl13b to mark primary cilia and DAPI to stain DNA.

## Kidney organoid differentiation

Organoids were differentiated in 384-well plates from human pluripotent stem cells (WTC11 iPS cells, Conklin lab, Gladstone Institute) that had been modified to disrupt *PKD2* . They were maintained feeder-free on 1% Reduced Growth Factor GelTrex (Life Technologies) in mTeSR1 (Stem Cell Technologies) and dissociated with Accutase (Stem Cell Technologies) with passage and procedure-matched controls. Identity of parental hPSC lines was confirmed to be correct based on matching the known morphology, karyotype, and pluripotency characteristics of these lines. Cell lines tested negative for mycoplasma. A total of 60,000 cells from each cell line were plated per well of a 24-well plate pre-coated with GelTrex in mTeSR1 supplemented with 10 µM Rho-kinase inhibitor Y27632 (StemGent). The media was replaced with 500 µl mTeSR1 + 1.5% GelTrex at 16 hr, 500 µl mTeSR1 at 36 hr, Advanced RPMI + Glutamax (from Life Technologies) + 12 µM CHIR99021 (Stemgent) at 50 hr, and RB (Advanced RPMI + Glutamax + B27 Supplement, from Life Technologies) at 86 hr. RB was changed 2 days later and every 3 days thereafter. Organoids typically became visible by light microscopy 12–18 days after plating.

## Cyst formation

In adherent cultures (untreated), 'cysts' were identified as large, balloon-like, translucent structures that swayed in response to agitation. Flat rings and dilated tubules were not counted as cysts and occasionally appeared even in non-PKD controls. Forskolin and 8-Br-cAMP (Sigma) were added to

adherent cultures on the 21st day of differentiation, resulting in rapid formation of cysts that typically did not sway in response to agitation. Wells were imaged using a Nikon TiE inverted wide-field microscope and cysts were quantified using ImageJ cell counter.

## Acknowledgements

We thank Scott Lab members for their valuable feedback on the manuscript. This work was supported by NIH: R01-DK119186 and 1R01DK119192-01 (JDS), NIH Awards T32 GM007270 (Gopalan), UG3TR002158 (Himmelfarb) and R01DK117914 (Freedman), F32DK121415 (Omar) and the Lara Nowak Macklin Research Fund.

## Additional information

### Funding

| Funder | Grant reference number | Author |
| --- | --- | --- |
| National Institutes of Health | R01-DK119186 | John D Scott |
| National Institute of Diabetes and Digestive and Kidney Diseases | 1R01DK119192-01 | John D Scott |
| National Institutes of Health | T32 GM007270 | Janani Gopalan |
| National Institute of Diabetes and Digestive and Kidney Diseases | F32DK121415 | Mitchell H Omar |
| University of Washington | Lara Nowak Macklin Research Fund | Mitchell H Omar |
| National Institutes of Health | UG3TR002158 | Jonathan Himmelfarb |
| National Institutes of Health | R01DK117914 | Benjamin S Freedman |

The funders had no role in study design, data collection and interpretation, or the decision to submit the work for publication.

### Author contributions

Janani Gopalan, Conceptualization, Data curation, Formal analysis, Funding acquisition, Validation, Investigation, Visualization, Methodology, Writing - original draft, Writing - review and editing; Mitchell H Omar, Validation, Investigation, Visualization, Writing - review and editing; Ankita Roy, Validation, Investigation; Nelly M Cruz, Katherine A Forbush, Resources; Jerome Falcone, Investigation; Kiana N Jones, Resources, Investigation; Jonathan Himmelfarb, Resources, Supervision; Benjamin S Freedman, Resources, Supervision, Methodology; John D Scott, Conceptualization, Resources, Supervision, Funding acquisition, Writing - review and editing

### Author ORCIDs

Janani Gopalan  https://orcid.org/0000-0001-7094-1074
Katherine A Forbush  http://orcid.org/0000-0003-4825-4766
John D Scott  https://orcid.org/0000-0002-0367-8146

### Decision letter and Author response

Decision letter https://doi.org/10.7554/eLife.67828.sa1
Author response https://doi.org/10.7554/eLife.67828.sa2

## Additional files

### Supplementary files

- Source data 1. Original blots included in the manuscript.
- Source data 2. Blots with legends added.

• Transparent reporting form

### Data availability

All data generated or analyzed during this study are included in the manuscript.

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
