## [Decision Letter]

**Acceptance summary:**

The initial observation that kicks off this interesting work, that AKAP220 loss of function increases ciliogenesis, is particularly dramatic. This result is important in itself, as negative regulators of ciliogenesis have been a subject of study, but mechanisms limiting ciliogenesis, particularly in vivo, are not well understood. The revised work bolsters the finding that AKAP220 controls HDAC6 function via PP1. This module, with PP1 as a structural component of a ternary complex, limits ciliation, at least in the collecting duct. Arriving at new insights into the regulators of ciliogenesis and ciliary PKD1/2 signaling in the kidney will be important both for understanding the variety of ciliary signaling modes, and for arriving at the biological insights that will underpin effective therapies for polycystic kidney disease.

**Decision letter after peer review:**

Thank you for submitting your article "Targeting an anchored phosphatase-deacetylase unit restores renal ciliary homeostasis" for consideration by *eLife*. Your article has been reviewed by 3 peer reviewers, one of whom is a member of our Board of Reviewing Editors, and the evaluation has been overseen by Philip Cole as the Senior Editor. The reviewers have opted to remain anonymous.

The reviewers have discussed their reviews with one another, and the Reviewing Editor has drafted this summary to help you prepare a revised submission. The reviewers were in consensus that this work may be suitable for publication with substantial revision. They appreciated the importance of the central finding that AKAP220 knockout collecting duct cells display increased ciliation. However, there were shared concerns that data in support of AKAP220 controlling HDAC6 function via PP1 were absent. In addition, key information regarding PLA, gene editing, and kidney samples, for example, was missing from the methods section. While there was appreciation of the insight that AKAP220 limits ciliation, the clinical and translational significance of these findings was, in their minds, unclear. Therefore, their consensus was that this work will require an extensive revision before publication. We have listed four key essential revisions, below, and other points from the reviewers are included as they will be of assistance in revising this work.

Essential Revisions (for the authors):

1) Provide evidence that AKAP220 interacts with HDAC6, directly or indirectly, using means complementary to PLA.

2) Demonstrate that HDAC6 fails to bind AKAP220-deltaPP1.

3) It should be noted that, as Tubacin is an anti-proliferative agent with some toxicity and HDAC6 may function independently of AKAP22, it remains unclear whether effects on cyst growth in vitro are mediated through the identified AKAP22-HDAC6 mechanism.

4) Add the requested information missing from the methods section.

*Reviewer #1 (Recommendations for the authors):*

In investigating the origins of increased ciliation, the authors detect reduced levels of the deacetylase in cells that lack AKAP220. At least by PLA, the authors detect AKAP220-HDAC6 complexes in cells and a role for AKAP220 in stabilizing HDAC6. The authors generate a mutant form of AKAP220, AKAP220-deltaPP1, that reduces HDAC6 activity and levels. The authors advance a model that AKAP220 interacts with PP1, which activates HDAC6 (through what they assert to be an "adaptor" non-enzymatic mechanism) to deacetylate tubulin and inhibit cilium biogenesis. Curiously, the authors do not reference Pugacheva et al., who first demonstrated roles for HDAC6 in negatively regulating cilium biogenesis. In support of this model, the authors observe no change in cilia biogenesis when HDAC6 is inhibited in AKAP220-ΔPP1 cells, suggesting that it may already be inhibited.

HDAC6 has previously been reported to deacetylate cortactin and promote cortical actin formation. At Figure 4, the authors switch to investigating dramatically increased nuclear acetylated cortactin and enhanced accumulation of cortical actin in AKAP220 mutant cells. Whether this increase in cortactin acetylation is mediated by HDAC6 is not tested.

Cortactin has previously been shown to be a negative regulator of cilium formation, but this work by Bershteyn et al. is also not referenced. As shown previously by still others, pharmacological modulation of actin can impact ciliation and pharmacological inhibition of cortical actin increases cilium formation. This result, confirmed by the authors, would seem to imply that the increased cortical actin observed in AKAP220 mutant cells is not sufficient to account for the increased ciliation. However, I did not read where the authors come to this conclusion.

In Figure 6, the authors switch to investigating the speed of trafficking ARL13B into cilia. AKAP220 mutant cells show long cilia. FRAP of the cilia tip for ARL13B shows increased recovery in AKAP220 mutant cells. The authors conclude that there may be increased trafficking of ARL13B from the cell body in these mutants. However, to make this inclusion, the authors should FRAP the entire cilium, not a portion of the cilium.

For Figure 7, the authors return to the subject of acetylation and show that an HDAC6 inhibitor decreases cyst formation by Pkd2-/- organoids. Whether this effect is due to increased ciliation, increased tubulin acetylation, or increased cortactin acetylation is not examined.

There is a lot of data in this work, but the central point of this paper that AKAP220 may physically interact with HDAC6 (directly or indirectly) and stabilizes it. (The effects of HDAC6 downregulation secondary to loss of AKAP220 are largely in support of Ran et al. and others previous findings that HDAC6 regulates ciliogenesis, and tubulin and cortactin acetylation.) The authors don't firmly establish the central point (AKAP220 interacts with and stabilizes HDAC6). For example, the authors conflate mutation of the PP1 binding site in AKAP220 with "disruption of HDAC6-PP1 attachment" without ever showing that HDAC6 fails to bind the mutant form of AKAP220, AKAP220-deltaPP1. Similarly, although depicted in models, the authors never establish whether this interaction is direct or indirect, or, if indirect as depicted, via PP1.

Other concerns regarding the experimental data are:

In Figure 1L, mutant and control conditions appear to be at different confluencies, a condition that is known to affect ciliogenesis. Differences in confluency are even more apparent in Figure 2A. The authors should control for cell confluence as this variable may be affecting ciliation.

Is AKAP220-deltaPP1 expressed at similar levels as AKAP220?

The authors claim that three-dimensional surface plots show that primary cilia morphology is altered in the AKAP220-ΔPP1 cells. However, it is not apparent from the image provided that this difference is consistent, there is no quantitation of this difference, and it is not clear that the imaging is of sufficient resolution to distinguish subtle differences in morphology.

In Figure 2, Western blot is used to argue that acetylated tubulin levels are affected by AKAP220 loss, whereas in Figure 3, imaging is used to argue that acetylated levels are affected by AKAP220 mutation. I suspect that the Western blot is more quantitative, so should be used in Figure 3. Regardless, consistency of approach would allow the reader to distinguish whether AKAP220 KO and deltaPP1 have similar or distinct effects on tubulin acetylation.

The effect of AKAP220 mutations on acetylated cortactin is impressive. Does AKAP220 affect total cortactin levels and distribution?

I'm unclear how, as reported in the text, 1.75 sec is 0.85 sec less than 1.6 sec (Figure 4R). Moreover, the text reports that the FRAP recovery t_1/2_ is 1.6 sec for wild type and 0.9 sec for AKAP220 KO cells. Based on Figure 4R, I think the text may have these values reversed.

The authors reveal that Cytochalasin D increased the percentage of ciliated wild type cells and then assert that, "Similar effects were observed in AKAP220KO cells (Figure 5G-L)." However, an increase in the AKAP220KO cells is not apparent and not quantified.

The authors hypothesize that AKAP220-associated PP1 promotes the formation of an actin barrier close to the basal body of the cilium. Is this actin barrier observed in the actin staining? Is this barrier disrupted in the mutant cells?

The authors assert, "HDAC6 mediated deacetylation of tubulin controls cilia depolymerization." However, the authors do not measure cilia depolymerization.

*Reviewer #2 (Recommendations for the authors):*

The study showcases a great number of different techniques, but in its current form the methods section is not sufficiently detailed. Specifically, technical details on the proximity ligation assay are absent, and little information is provided concerning gene editing approaches including establishing the AKAP220deltaPP1 mIMCD3 cell line, and this would need to be improved.

*Reviewer #3 (Recommendations for the authors):*

1. The translational angle significantly weakens the manuscript as currently written. Both the introduction and an important part of the discussion are focused on the relevance of this study's findings to PKD, and yet the one result supporting this is inconsistent with their model (that inactivation of the AKAP220/HDAC6 pathway enhances ciliogenesis and ciliary function). At a minimum, the authors should propose an explanation for how blocking ciliogenesis in an ADPKD model attenuates disease whereas in their studies promoting ciliogenesis achieves the same effect. Alternatively, the authors might consider dropping the last experiment and refocus the study on its relevance to basic ciliary biology, or add more studies to more convincingly show the relevance of their pathway to modulating cyst growth (genetic studies in organoids and in vivo). If they pursue the latter option, they still should provide an explanation for how boosting ciliogenesis can improve disease whereas the published studies suggest the opposite.

2. The study by Sun et al., is not accurately described in the manuscript. The authors cite Sun et al. as evidence that aberrant HDAC6 activity has been implicated in cyst growth. This is not the case---the authors used a broad, multi-class HDAC inhibitor in their studies and never specifically reference HDAC6 as the likely target.

3. Some of the methods are not well described. For example, we are not told how many kidney samples, from how many mice, at what age mutants and controls were studied, and how fields were selected for analysis. For the organoid studies, they should describe the age of the organoids prior to initiating treatments and how fields were selected for quantitation given the known variability in the phenotype.

4. It appears that the authors accidentally flipped the results in their description of the FRAPing studies on pages 17-18. Both the data in Figure 4 and the movies show that the WT sections recover more quickly than do the mutant ones but the text describes the opposite.

---

## [Author Response]

Essential Revisions (for the authors):1) Provide evidence that AKAP220 interacts with HDAC6, directly or indirectly, using means complementary to PLA.2) Demonstrate that HDAC6 fails to bind AKAP220-deltaPP1.

Points 1 and 2 can be addressed together. Our response is in three parts:

a) New biochemical data in figure 3D (lane 1) shows that immunoprecipitation of HDAC6 from mIMCD3 cells results in co-fractionation of AKAP220. Importantly, co-fractionation of the anchoring protein is markedly reduced when experiments are performed in cells expressing AKAP220-∆PP1 (Figure 3D, lane 2). Immunoblot detection of AKAP220 (top), HDAC6 (mid) and GAPDH loading controls (bottom) reveal equivalent levels of all proteins in mIMCD3 cell lysates (Figure 3D). These new data indicate that removal of the principle PP1-interaction motif on AKAP220 negatively impacts association with HDAC6. This is compatible with data from Shenolikar and colleagues reporting that PP1 directly interacts with HDAC6 (Brush et al., 2004). Hence, loss of PP1-anchoring uncouples HDAC6 association with AKAP220 signaling islands.

However, our findings are equally consistent with the notion that removal of the KVXF motif on AKAP220 impedes, but does not fully abolish, PP1-anchoring. As mentioned in the original text we previously identified three secondary PP1-interactive surfaces on AKAP220 (Schillace and Scott, 1999; Schillace et al., 2001). Our 2001 report was a seminal observation of multiple contacts between PP1 and its targeting subunits. Subsequently the notion of multisite contact between serine/threonine phosphatases and their binding partners has become a standard view (Hoermann et al., 2020).

Since it is not technically feasible to remove all four PP1-interactive surfaces on AKAP220 and express a functional protein, we can only conclude that there is a reduced association of the PP1-HDAC6 sub complex in cells expressing AKAP220-∆PP1. Two plausible mechanisms support this conclusion. First, removal of the primary PP1-targeting motif on AKAP220 lessens binding of the PP1-HDAC6 subcomplex. Second, mutation of the KVXF motif could expose secondary HDAC6 binding determinants that could reside within AKAP220 or any of its other binding partners. Irrespective of either mechanism, we have definitively shown that HDAC6 interacts with AKAP220. Moreover, loss of PP1-anchoring significantly impacts maintenance of HDAC6 within AKAP220 signaling islands. These new results are introduced on page 14, line 18 of the results and expanded upon on page 34, line 20 in the discussion.

b) HDAC6 is a phosphoprotein. The AKAP220-associated kinase GSK3 phosporylates HDAC6 on serine 22 (Chen et al., 2010). Thus recruitment into AKAP220 signaling islands would optimally position HDAC6 for control by reversible phosphorylation. With this in mind (and in response to reviewer comments) we monitored the phosphoryaltion status of Ser22 on HDAC6 in wildtype and AKAP220-∆PP1 cells (Figure 3—figure supplement 2A and B). Immunoblot detection of pSer22 HDAC6 was robust in wildtype mIMCD3 cell lysates (Figure 3—figure supplement 2A, upper panel, lane 1). Importantly, p-HDAC6 signal is reduced in cell lysates from AKAP220-∆PP1 IMCD3 cells (Figure 3—figure supplement 2A, upper panel, lane 2). Control experiments confirm equivalent amounts of total HDAC6 in cell lysates from both genotypes (Figure 3—figure supplement 2B, upper panel). Thus phospho-HDAC6 levels are reduced in cells engineered to express AKAP220-∆PP1. This result is consistant with dissociation of HDAC6 from the anchored kinase-phosphatase signaling complex in a manner that promotes a concomitant loss of regulation by protein phosporylation. These new data are intorduced on page 15, line 3.

c) Immunofluorescence detection of phospho Ser 22-HDAC6 provided independent confirmation of this later notion (Figure 3—figure supplement 2C-I). mIMCD3 cells were stained for p-HDAC6 (red), AKAP220 (green). Acetyl tubulin (magenta) was used as a ciliary marker and DAPI (blue) marked nuclei (Figure 3—figure supplement 2C and F). In wildtype cells, co-distribution of AKAP220 and p-HDAC6 was evident at the base of cilia (Figure 3—figure supplement 2D, arrow). In contrast, p-HDAC6 signals were reduced in AKAP220-∆PP1 cells (Figure 3—figure supplement 2G, arrow). Quantification of p-HDAC6 levels at the base of cilia in multiple cells is presented Figure 3—figure supplement 2I. Please note antibody compatabilty issues prevented us from measuring total HDAC6 in situ.

The new data shows that HDAC6 is a component of AKAP220 signaling islands by two methods complimentary to PLA. Also, HDAC6 binding is reduced in mIMCD3 cells engineered to express AKAP220-∆PP1. This information has been incorporated into Figure 3D, and Figure 3—figure supplement 2A-I. The results are discussed in detail beginning on page 34, line 20.

3) It should be noted that, as Tubacin is an anti-proliferative agent with some toxicity and HDAC6 may function independently of AKAP22, it remains unclear whether effects on cyst growth in vitro are mediated through the identified AKAP22-HDAC6 mechanism.

This is a valid point. HDAC6 modulates a variety of cellular functions including control of cell morphology, cell adhesion and migration and tumor metastasis. Interestingly these are known sites of action of the AKAP220 signaling complexes (Boyault et al., 2007; Logue et al., 2011; Wong and Scott, 2004). Thus, it is reasonable to speculate that tubacin could target the AKAP220-PP1-HDAC6 axis in these cellular contexts.

These points notwithstanding, the reviewers are correct in asking us to acknowledge that HDAC6 also modulates a variety of cellular processes independently of the anchoring protein. These are likely to include control of immune cell interactions and modulation of the unfolded protein response (Hideshima et al., 2016). We hope that qualifying our conclusions by stating that the AKAP220-PP1-HDAC6 axis is a new, but not the only molecular target for Tubacin satisfactorily responds to the reviewer concerns. Also, we now state that tubacin action at multiple sites could impact the anti-proliferative effect on cyst formation. These statements are introduced on page 31, line 7 of the results and expanded upon on page 37, line 6 of the discussion.

4) Add the requested information missing from the methods section.

We have added the requested information in the methods, in particular details about the proximity ligation assays, cell lines generated by CRISPR Cas9 gene editing, production of the mouse knock out and culturing of iPSC organoids.

Reviewer #1 (Recommendations for the authors):In investigating the origins of increased ciliation, the authors detect reduced levels of the deacetylase in cells that lack AKAP220. At least by PLA, the authors detect AKAP220-HDAC6 complexes in cells and a role for AKAP220 in stabilizing HDAC6. The authors generate a mutant form of AKAP220, AKAP220-deltaPP1, that reduces HDAC6 activity and levels. The authors advance a model that AKAP220 interacts with PP1, which activates HDAC6 (through what they assert to be an "adaptor" non-enzymatic mechanism) to deacetylate tubulin and inhibit cilium biogenesis. Curiously, the authors do not reference Pugacheva et al., who first demonstrated roles for HDAC6 in negatively regulating cilium biogenesis. In support of this model, the authors observe no change in cilia biogenesis when HDAC6 is inhibited in AKAP220-ΔPP1 cells, suggesting that it may already be inhibited.HDAC6 has previously been reported to deacetylate cortactin and promote cortical actin formation. At Figure 4, the authors switch to investigating dramatically increased nuclear acetylated cortactin and enhanced accumulation of cortical actin in AKAP220 mutant cells. Whether this increase in cortactin acetylation is mediated by HDAC6 is not tested.

We have performed experiments to test if inhibiting HDAC6 influences cortactin acetylation. These data are included for the reviewer to evaluate in Author response image 1. The overall conclusion is that tubacin treatment enhances acetylated cortactin in all three cell types.

**Author response image 1. respfig1:** Inhibition of HDAC6 enhances acetylated cortactin in mIMCD3 cells. Immunofluorescence images of acetylated cortactin in DMSO or Tubacin-treated (A and B) wildtype, C and (D) AKAP220KO and (E and F) AKAP220-ΔPP1 mIMCD3 cells. Insets depict marked nucleus (yellow box, DMSO); (white box, tubacin-treated). Each image and the corresponding 3D surface plots show changes in protein level. (G and H) Quantification of integrated density shows amalgamated data (20 nuclei) from wildtype (black), AKAP220KO (gray), and AKAP220-ΔPP1 (coral) cells.

Cortactin has previously been shown to be a negative regulator of cilium formation, but this work by Bershteyn et al. is also not referenced. As shown previously by still others, pharmacological modulation of actin can impact ciliation and pharmacological inhibition of cortical actin increases cilium formation. This result, confirmed by the authors, would seem to imply that the increased cortical actin observed in AKAP220 mutant cells is not sufficient to account for the increased ciliation. However, I did not read where the authors come to this conclusion.

Although we have observed redistribution of cortical actin in AKAP220KO and AKAP220-∆PP1 cells as compared to wildtype cells, there is no difference in the total amount of actin between the three cell types. This leads us to conclude that altered distribution of actin (potentially loss of actin at the base of the cilium) could account for enhanced ciliation. As per the reviewer’s suggestion, we have now added a sentence about cortactin antagonizing cilium formation in our discussion and cited Bershteyn et al., 2010. This appears on page 36, line 3.

In Figure 6, the authors switch to investigating the speed of trafficking ARL13B into cilia. AKAP220 mutant cells show long cilia. FRAP of the cilia tip for ARL13B shows increased recovery in AKAP220 mutant cells. The authors conclude that there may be increased trafficking of ARL13B from the cell body in these mutants. However, to make this inclusion, the authors should FRAP the entire cilium, not a portion of the cilium.

We understand this reviewer’s issue and originally considered photobleaching the entire cilium. The primary cilium is a microscopic structure. Thus, a recognized to caveat of this approach is that photobleaching the entire cilium can negatively affect the base of the organelle (Blasius et al., 2019). To limit damage, we chose to selectively photobleach the tips of cilia. This method also allowed us to target only the region of interest and ensure that we were measuring protein recovery at the correct location. Another advantage to this method is that it by-passes artifacts that can be introduced because extended regions of the primary cilia can drift in and out of focus during the time course of the experiment. We have clarified this point on page 27, line 18.

For Figure 7, the authors return to the subject of acetylation and show that an HDAC6 inhibitor decreases cyst formation by Pkd2-/- organoids. Whether this effect is due to increased ciliation, increased tubulin acetylation, or increased cortactin acetylation is not examined.

This is a valid point. We are exploring this matter further. This new line of inquiry is out with the scope of the current study. We have modified the discussion on page 37, line 6 to reflect these concerns.

There is a lot of data in this work, but the central point of this paper that AKAP220 may physically interact with HDAC6 (directly or indirectly) and stabilizes it. (The effects of HDAC6 downregulation secondary to loss of AKAP220 are largely in support of Ran et al. and others previous findings that HDAC6 regulates ciliogenesis, and tubulin and cortactin acetylation.) The authors don't firmly establish the central point (AKAP220 interacts with and stabilizes HDAC6). For example, the authors conflate mutation of the PP1 binding site in AKAP220 with "disruption of HDAC6-PP1 attachment" without ever showing that HDAC6 fails to bind the mutant form of AKAP220, AKAP220-deltaPP1. Similarly, although depicted in models, the authors never establish whether this interaction is direct or indirect, or, if indirect as depicted, via PP1.

This important issue fully addressed in the response to essential revisions 1 and 2.

Other concerns regarding the experimental data are:In Figure 1L, mutant and control conditions appear to be at different confluencies, a condition that is known to affect ciliogenesis. Differences in confluency are even more apparent in Figure 2A. The authors should control for cell confluence as this variable may be affecting ciliation.

The reviewer raises an important point about cell confluence affecting ciliation. The three cell types used in the study were consistently plated at the same density and treated with drugs/imaged under the same conditions. We noted early on that the AKAP220KO or AKAP220-∆PP1 appear less confluent and cover the entire coverslip faster than wildtype. This phenomenon appears to involve differential coupling to the integrins and cadherins that is the topic a separate manuscript currently in preparation.

Is AKAP220-deltaPP1 expressed at similar levels as AKAP220?

New data presented Figure 3D shows that AKAP220 and AKAP220-deltaPP1 are expressed at comparable levels. This is discussed on page 14, line 18.

The authors claim that three-dimensional surface plots show that primary cilia morphology is altered in the AKAP220-ΔPP1 cells. However, it is not apparent from the image provided that this difference is consistent, there is no quantitation of this difference, and it is not clear that the imaging is of sufficient resolution to distinguish subtle differences in morphology.

We recognize that the 3D plots are challenging to interpret. Yet, differences in height and width of the plotted data between wildtype and AKAP220-deltaPP1 concisely indicate the morphological differences between two cell types. We now emphasize this point on page 16, line 6.

In Figure 2, Western blot is used to argue that acetylated tubulin levels are affected by AKAP220 loss, whereas in Figure 3, imaging is used to argue that acetylated levels are affected by AKAP220 mutation. I suspect that the Western blot is more quantitative, so should be used in Figure 3. Regardless, consistency of approach would allow the reader to distinguish whether AKAP220 KO and deltaPP1 have similar or distinct effects on tubulin acetylation.The effect of AKAP220 mutations on acetylated cortactin is impressive. Does AKAP220 affect total cortactin levels and distribution?

Cortactin distribution is also altered in the AKAP220 mutant cells. This is now mentioned in the text on page 20, line 21.

I'm unclear how, as reported in the text, 1.75 sec is 0.85 sec less than 1.6 sec (Figure 4R). Moreover, the text reports that the FRAP recovery t_1/2_ is 1.6 sec for wild type and 0.9 sec for AKAP220 KO cells. Based on Figure 4R, I think the text may have these values reversed.

We thank the reviewer for catching this. The word “reduction” should have been “increase”. We have changed the text.

The authors reveal that Cytochalasin D increased the percentage of ciliated wild type cells and then assert that, "Similar effects were observed in AKAP220KO cells (Figure 5G-L)." However, an increase in the AKAP220KO cells is not apparent and not quantified.

While cytochalasin D treatment led to a modest increase in AKAP220KO and AKAP220-∆PP1 cilia numbers, these cells displayed developed longer cilia upon drug treatment (Figure 6G-M and Figure 6—figure supplement 1). This leads us to consider if actin depolymerization also elongates cilia by allowing proteins to move freely into the organelle. We have altered the text on page 24, line 12 to reflect the difference between the cell types more accurately.

The authors hypothesize that AKAP220-associated PP1 promotes the formation of an actin barrier close to the basal body of the cilium. Is this actin barrier observed in the actin staining? Is this barrier disrupted in the mutant cells?

Visualizing the actin barrier at the base of the cilium has been a challenge as actin is so prevalent in the cell. However, we note of actin redistribution is altered in the mutants, there is slower recovery of actin after photobleaching and ARL13b tracks more rapidly into mutant cilia. Taken together these observations allow us to postulate that changes in actin at the base of cilia may contribute to the extended cilia phenotypes that we are reporting. This is something we are interested in pursuing as a future line of investigation.

The authors assert, "HDAC6 mediated deacetylation of tubulin controls cilia depolymerization." However, the authors do not measure cilia depolymerization.

With all due respect to reviewer 1 we cannot find this statement in the text or figure legends.

Reviewer #2 (Recommendations for the authors):The study showcases a great number of different techniques, but in its current form the methods section is not sufficiently detailed. Specifically, technical details on the proximity ligation assay are absent, and little information is provided concerning gene editing approaches including establishing the AKAP220deltaPP1 mIMCD3 cell line, and this would need to be improved.

Thank you for bringing this up. We have improved the ‘methods’ section. (See response 4 to the essential revisions) The expanded methods section begins on page 53.

Reviewer #3 (Recommendations for the authors):1. The translational angle significantly weakens the manuscript as currently written. Both the introduction and an important part of the discussion are focused on the relevance of this study's findings to PKD, and yet the one result supporting this is inconsistent with their model (that inactivation of the AKAP220/HDAC6 pathway enhances ciliogenesis and ciliary function). At a minimum, the authors should propose an explanation for how blocking ciliogenesis in an ADPKD model attenuates disease whereas in their studies promoting ciliogenesis achieves the same effect. Alternatively, the authors might consider dropping the last experiment and refocus the study on its relevance to basic ciliary biology, or add more studies to more convincingly show the relevance of their pathway to modulating cyst growth (genetic studies in organoids and in vivo). If they pursue the latter option, they still should provide an explanation for how boosting ciliogenesis can improve disease whereas the published studies suggest the opposite.

We respect the reviewer’s opinion that **“**translational angle significantly weakens the manuscript” and have softened aspects of the abstract, introduction and discussion (see public response to reviewer 3). These sections have been amended to deemphasize a link to Autosomal dominant polycystic kidney disease. As mandated in the essential revisions we now state that tubacin is an anti-proliferative agent with some toxicity. We further acknowledge that pools of HDAC6 not associated with AKAP220 will be suspectable to the drug. Thus, targeting the AKAP220-PP1-HDAC6 axis may represent one of several pharmacological effects associated with this drug. We hope that these changes beginning on line 1 of the abstract, page 3, line 10 of the introduction, page 31, line 7 of the results and page 37, line 6 of the discussion reflect reviewer 3’s concern that we over emphasized the “translational aspects “of our study.

2. The study by Sun et al., is not accurately described in the manuscript. The authors cite Sun et al. as evidence that aberrant HDAC6 activity has been implicated in cyst growth. This is not the case---the authors used a broad, multi-class HDAC inhibitor in their studies and never specifically reference HDAC6 as the likely target.

We have now included a more appropriate reference that specifically substantiates our point about aberrant activity HDAC6 in cyst growth.

3. Some of the methods are not well described. For example, we are not told how many kidney samples, from how many mice, at what age mutants and controls were studied, and how fields were selected for analysis. For the organoid studies, they should describe the age of the organoids prior to initiating treatments and how fields were selected for quantitation given the known variability in the phenotype.

We have now added more details about our mice and organoid experiments to bolster the methods section beginning on page 53.

4. It appears that the authors accidentally flipped the results in their description of the FRAPing studies on pages 17-18. Both the data in Figure 4 and the movies show that the WT sections recover more quickly than do the mutant ones but the text describes the opposite.

We thank the reviewer for catching this. The word “reduction” should have been “increase”. This has been corrected in the text.